

# From ionising radiation to air ion formation in the lower atmosphere

Xuemeng Chen[1*], Veli-Matti Kerminen[1], Jussi Paatero[2], Pauli Paasonen[1], Hanna E. Manninen[1], Tuomo Nieminen[1,3], Tuukka Petäjä[1] and Markku Kulmala[1]

5   [1]Department of Physics, University of Helsinki, P.O. Box 64, FI-00014 Helsinki, Finland
[2]Finnish Meteorological Institute, P.O. Box 503, FI-00101 Helsinki, Finland
[3]Department of Applied Physics, University of Eastern Finland, P.O. Box 1627, FI-70211 Kuopio, Finland

10  *Correspondence to*: Xuemeng Chen (xuemeng.chen@helsinki.fi)





**Abstract.** Most of the ion production in the atmosphere is attributed to ionising radiation. In the lower
atmosphere, ionising radiation consists mainly of the decay emissions of radon and its progeny, gamma
radiation of the terrestrial origin as well as photons and elementary particles of cosmic radiation. These
types of radiation produce ion pairs via the ionisation of nitrogen and oxygen as well as trace species in
the atmosphere, the rate of which is defined as the ionising capacity. Larger air ions are produced out of
the initial charge carriers by processes, such as clustering or attachment to pre-existing aerosol particles.
This study aimed 1) to identify the key factors responsible for the variability in ionising radiation and in
the observed air ion concentrations, 2) to reveal the linkage between them, and 3) to provide an in-depth
analysis into the effects of ionising radiation on air ion formation, based on measurement data collected
during 2003-2006 from a boreal forest site in southern Finland. In general, gamma radiation dominated
the ion production in the lower atmosphere. Variations in the ionising capacity came from mixing layer
dynamics, soil type and moisture content, meteorological conditions, long-distance transportation, snow
cover attenuation and precipitation. Similar diurnal patterns to variations in the ionising capacity were
observed in air ion concentrations of the cluster size (0.8-1.7 nm in mobility diameters). Clear
promotion effects of the ionising radiation on air ion production were demonstrated. Typically, features
observed in the 0.8-1 nm ion concentration were in connection to variations of the ionising capacity.
Further by carefully constraining perturbing variables, a clear relationship was also identifiable between
the cluster ion concentration and the ionising capacity, proving the functionality of ionising radiation in
air ion production in the lower atmosphere.

**1. Introduction**

Ambient radioactivity in the lower atmosphere supplies ionising energy for the production of electric
charges in the air. It consists of natural and anthropogenic radioactivity. The anthropogenic fraction
comes mainly from routine and accidental emissions from nuclear power plants and related facilities as
well as nuclear detonations. Minor emissions of natural radioactivity occur also in connection with
various mining activities. Natural radioactivity is composed of the decay emissions of naturally
occurring radionuclides and cosmic radiation. Alpha and beta particles as well as the associated
gamma and x-ray photons constitute the ionising radiation from natural radionuclides. Radon-222





is a naturally occurring radioactive gas, which is the daughter nuclide of radium-226 that is typically present in soil grains. Once formed, radon can diffuse through soil pores and eventually enter into the atmosphere (Chen et al., 2016;Nazaroff, 1992). Radon and its progeny undergo alpha and beta decays in the atmosphere and, together with the related gamma radiation, supply the energy for ionisation. Apart

from radon, also gamma radiation from the Earths' crust as well as photons and elementary particles of cosmic radiation contribute to the creation of electric charges in the lower atmosphere. Typically, 32.5-35 eV is needed to produce an ion pair in the atmosphere (Krause et al., 2002), with an average expenditure of 34 eV per ion pair often accepted in the lower atmosphere (Wilkening, 1981;Jesse and Sadaukis, 1957;Laakso et al., 2004).

Due to the atmospheric abundance of nitrogen ($N_2$) and oxygen ($O_2$), the generated electric charges from the ionisation process are initially carried by the derivatives of them. These initial charge carriers are known as primary ions, which are simple molecular ions. Primary ions are consumed either directly or via the formation of more complex molecular ions by 1) ion-ion recombination, 2) clustering, 3)

charge transfer to pre-existing aerosol particles (or clusters) or gaseous species in the atmosphere and 4) sink to foreign surfaces. These processes may involve both chemical reactions and physical transformations. The recombination and sink to foreign surfaces lead to a reduced amount of electric charges in the air, whereas the clustering and charge transfer result in either charged or neutral gaseous species, clusters and aerosol particles. The term 'air ion' refers to all airborne substances that are

electrically charged, ranging from primary ions to charged aerosol particles. The relationship between primary ions, molecular ions and cluster ions are illustrated in Fig. 1. At present, the number size distribution of air ions can be measured down to about 0.8 nm in Millikan mobility size by using ion spectrometers (Tammet, 2011;Manninen et al., 2009;Tammet, 2006). Most air ions concentrate at the lowest size band with a diameter of 0.8-1.7 nm (Manninen et al., 2009), which is generally known as

the cluster size range (Tammet, 2012, 1995). In principle, this size range contains large molecular ions and clusters of molecular ions. Ions smaller than 0.8 nm comprise mostly relatively simple molecular ions, which are either primary ions or originate from the survived fraction of primary ions from recombination. The critical cluster size was found to be 1.5±0.3 nm for atmospheric nucleation





(Kulmala et al., 2013). Once the critical cluster size is reached, further growth of a cluster in size is energetically favoured (Vehkamäki, 2006). Larger air ions than 1.7 nm are therefore typically viewed as nanoparticles, observable especially during new particle formation (NPF) events (Manninen et al., 2009;Kulmala et al., 2012).  Since there is no clear size separation between molecular ions and cluster

ions, we hereafter refer ions in the 0.8-1.7 nm size range as cluster ions, unless otherwise mentioned.

Air ions were historically concerned in the discipline of atmospheric electricity (Israël, 1970), because their flow in the electric field of the earth-atmosphere system serves as the measureable conduction current in the atmosphere (Wilson, 1921;Tinsley, 2008;Harrison and Carslaw, 2003). The ionisation

process gives rise to the production of primary ions (e.g. (Tinsley, 2008;Harrison and Carslaw, 2003;Israël, 1970)). Although the connection of primary ion formation to ionising radiation is self-evident, except for a few attempts (Hirsikko et al., 2007;Laakso et al., 2004), minor efforts have been invested in understanding the causality between ionising radiation and observed air ions in the lower atmosphere. Moreover, there is a lack of quantification on the underlying processes. Such deficiencies

prompt the motivation of this work to examine the connection between ionising radiation and air ion formation, aiming 1) to identify the key factors responsible for the variability in ionising radiation and in the observed air ion concentrations, 2) to reveal the linkage between the observed air ions and ionising radiation and 3) to provide an in-depth analysis into the effects of ionising radiation on air ion formation. We will first introduce factors that cause the seasonal and diurnal variability in ionising

radiation and air ions, and then exposit the relationship between ionising radiation and air ion formation.

## 2. Materials and methods

The data presented in this work were collected from a boreal forest site, which is known as the SMEAR II station located at Hyytiälä, Southern Finland (61°51´N, 24°17´E, 181 m above sea level) (Hari and

Kulmala, 2005), during 2003-2006. The monitoring devices of the ionising radiation have been deployed to this site since 2000 and the air ion measurement was initialised in 2003. Both of these measurements have been integrated into the long-term monitoring system on the site. Other data used in



this work included the condensation sink (CS (Kulmala et al., 2012)) derived from the number size distribution of ambient aerosol particles, ambient relative humidity (RH), soil water content (SWC), soil temperature, snow cover depth and modelled mixing layer height (MLH). The estimates of MLH were retrieved from the European Centre for Medium-Range Weather Forecasts (ECMWF) Meteorological

Archival and Retrieval System (MARS), Reading, UK (www.ecmwf.int).

## 2.1. Instrumentation and data processing

The ionising radiation measurement system consists of a radon monitor and a gamma spectrometer. Both devices are maintained by the Finnish Meteorological Institute (FMI). The air ion data was obtained by a Balanced Scanning Mobility Analyser (BSMA) (Tammet, 2006), which is part of the

aerosol monitoring system at the station. A Differential Mobility Particle Sizer (DMPS) has been responsible for observing the ambient aerosol number size distribution on this site since 1996. Data are presented in local winter time (UTC+2:00).

### 2.1.1. Ionising radiation measurement

The radon monitor is a dual fixed filter-based instrument and measures the aerosol beta activity. Its inlet

is fixed at 6 m above the ground and this device has been described in detail by Paatero et al. (1994). Briefly, it is made up of two cylindrical Geiger-Müller counters covered with glass-fibre filters in lead shielding. A pump controller directs the airflow to either counter for a 4-h period alternatively, allowing the beta activity on the other counter to decay. The counting efficiency for beta particles is determined by the geometric configuration of the counting system together with the intrinsic detection efficiency of

the GM tubes, which is 0.96% and 4.3% for $^{214}$Pb and $^{214}$Bi beta particles, respectively. The cumulative counts are logged at 10-min intervals. The re-establishment of the radon activity concentration into Bq m$^{-3}$ from the recorded count rates was achieved following the method and assumption given by Paatero et al. (1994) and Chen et al. (2016).

The gamma spectrometer is a scintillation-type detector using a 76 mm × 76 mm NaI(Tl) as the

detection medium (Laakso et al., 2004;Hirsikko et al., 2007), which is kept at a height of 1.5 m above




the ground. Pulse height spectra over the energy range of 100-3000 keV are measured with a multichannel analyser and the total counts within the energy range are recorded with a time resolution of 10 min. The total gain of the detecting system is kept constant via digital spectrum stabilization using the potassium-40 gamma peak (1460 keV) as the reference. The recorded count rates were converted

into dose rates in the air (μSv/h) by a calibration factor obtained from an instrumental comparison to a pressurised ionisation chamber. For the determination of the ionising capacity in this work, the total gamma dose rate data was used instead of the spectral information.

### 2.1.2. Air ion measurement

The most state-of-the-art measurement techniques nowadays, usually employing a mass spectrometer as

the detector, such as the Atmospheric Pressure Interface Time- of-Flight Mass Spectrometer (APi-TOF) (Junninen et al., 2010;Junninen et al., 2016), are able to track air ions down to molecular sizes and characterise their chemical composition, but are lacking  the capability for providing information on the number concentration. Ion spectrometers are the most deployed type of devices for the study of air ion concentrations in segregated mobility-channels (Hirsikko et al., 2011). The Balanced Scanning Mobility

Analyser (BSMA) (Tammet, 2006) and the Neutral Cluster and Air Ion Spectrometer (NAIS) (Mirme and Mirme, 2013;Kulmala et al., 2007;Manninen et al., 2016) are examples of this category. Such instruments, however, are restricted by their upper and lower measurement limits of mobility. Currently, the highest measurable mobility with ion spectrometers is 3.2 cm$^2$/Vs (Tammet, 2006, 2011;Mirme and Mirme, 2013), equivalent to a mobility diameter of about 0.8 nm based on the Stokes-

Millikan equation. The mobility size of a particle may vary with the choice of a mobility-diameter model as well as parameters and coefficients used therein (e.g. Mikael et al., 2011;Tammet, 1995, 2011;Mäkelä et al., 1996;Fernandez-Garcia and Fernandez de la Mora, 2013).

Ambient air ion data used in this work were measured by a BSMA. The inlet of this instrument was at a

height of 1.5 m above the ground. The BSMA is an integral-type counter (Tammet, 1970), which offers air ion mobility spectra by a continuous voltage-scanning system. It is composed of two plain aspiration condensers, one for each polarity. An electrofilter is installed at the inlet of each condenser. The BSMA




has a total flow rate of 2640 l/min, which is split into two streams at the inlet, supplying each condenser with a flow of 1320 litre per minute. For either condenser, only 1/8 of the air stream passing through the electrofilter contains air ions, which is taken as the sample flow, leaving the rest ion-free fraction as the sheath flow. After the electrofilter, there exists an electric field in each condenser, created between a

grounded collecting electrode and a repelling electrode that is kept at a high potential, where the trajectories of sampled air ions are deflected towards the collecting electrode. A 33-mm long and 170-mm wide sensing unit is inlaid on the counter electrode of each polarity. Both sensing units share a common electrometer for counting the captured ions. A bridging circuit balances the voltage supply onto the repelling electrodes, so that the induced electric currents on the collectors during the voltage

scanning are equal in magnitudes, but of opposite polarity. This design avoids the generation of noise in the common electrometer. The BSMA segregates air ions based on their mobility into 16 channels in the range of 0.032-3.2 $cm^2/Vs$. The air ion number concentration measured in the mobility domain was automatically processed into number size distributions by the recording programme, with the particle size expressed in Tammet's mass diameter (Tammet, 1995, 2006), which was subsequently converted to

the mobility diameter based on the mobility-size conversion model proposed by Tammet (1995) and the Stokes-Millikan equation. The core of the Tammet model lies on a modified Millikan equation, which approaches the Chapman-Enskog equation in the free molecular scale and the Millikan equation in the macroscopic scale. The equivalent mobility diameter range of the BSMA is 0.8-8 nm. In this work, particle sizes are always presented in mobility diameters.

**2.1.3. Other instrumentation**

A Differential Mobility Particle Sizer (DMPS) gives the information on the aerosol number size distribution (Wiedensohler et al., 2012). At the SMEAR II station, a twin-DMPS system is deployed, with one responsible for the mobility size range of 3-50 nm and the other for larger sizes (Kulmala et al., 2012;Aalto et al., 2001). Each DMPS consists primarily of a differential mobility analyser (DMA)

and a condensation particle counter (CPC). The sample air passes through a common bipolar diffusion charger, where aerosol particles are brought to a thermal charge equilibrium. Subsequently, the sample stream is divided into two to be directed to individual DMPSs, where aerosol particles are size





segregated in the DMA by changing the voltage step-wisely and then counted in the CPC. The DMPS covered 3-500 nm until December 2004, after which the size range was expanded to 3-1000 nm. The determination of the condensation sink (CS) was conducted following the method presented by Kulmala et al. (2012).

The snow cover depth was measured manually on a weekly basis on seven different locations at the SMEAR II station. The soil temperature was measured with a silicon temperature sensor (Philips KTY81-110) and the soil volumetric water content with a Time-Domain Reflectometry (TDR). The soil data were spatially averaged from five measurement locations, which were available for the

organic (O) horizon (5 cm depth, above the mineral layer), A horizon (0-5 cm from the mineral soil surface), B horizon (5-23 cm depth) and C horizon (23-60 cm depth) (Pumpanen et al., 2003) . Only the O horizon data were used in this work.

The ambient relative humidity (RH) was calculated from the measured dew point temperature and air

temperature on the mast at the height of 16 m. The dew point was measured with a chilled mirror sensor (DewTrak Model 200M Meteorological Humidity System, EdgeTech, Marlborough, MA, USA), which had an accuracy of ± 0.3 °C. The air temperature was obtained with a PT-100 sensor, which was protected from solar radiation and ventilated by a fan. The accuracy of this sensor was determined to be ± 0.2 °C, via a comparison with an Hg-thermometer.

**2.2. Ionising capacity**

The ionising capacity ($Q$) is defined as the potential amount of ion pairs produced per unit time upon ionisation by ionising radiation in the atmosphere. The ionising capacity was determined on the basis of the assumption that 34 eV is needed for the production of an ion pair in the air. The energy released by radon-222 and its short-lived progeny during alpha and beta decay were taken into consideration. The

decay mode and energy accounted for the ionising capacity conversion are listed in Table 1. The minor gamma fraction from the radon-222 decay was assumed to be detectable by the gamma spectrometer. Therefore, the data obtained by the gamma spectrometer could be considered to represent the total





gamma radiation, including the terrestrial fraction, the cosmic fraction and the fraction from radon decay. The conversion from the activity concentration (Bq m$^{-3}$) to the ionising capacity (cm$^{-3}$ s$^{-1}$) is straightforward, when the amount of energy released during the radioactive decay is known. The conversion from total gamma dose rate (DR, μSv h$^{-1}$) to the ionising capacity (cm$^{-3}$ s$^{-1}$) is described by

the following equation,

$$Q_\gamma = \frac{DR \cdot \rho_{air}}{W},$$  (1)

where $W$ is the amount of energy needed for the generation of an ion pair in the air (assumed to be 34 eV) and $\rho_{air}$ is the density of air, which can be derived from the ideal gas law.

For conciseness and clarity, hereafter the ionising capacity results from the alpha and beta decay of radon-222 and its short-lived daughter nuclides is denoted as the radon ionising capacity ($Q_{Rn}$) and the ionising capacity from total gamma radiation as the gamma ionising capacity ($Q_\gamma$).

### 3. Results and discussions

#### 3.1. Seasonal and diurnal patterns in the natural ionising capacity

The natural ionising capacity has generally the same dynamical variations as ionising radiation, from which the ionising capacity was derived. A decline in the gamma ionising capacity was seen in the seasonal profile prior to the lowest value (4.5 cm$^{-3}$ s$^{-1}$) reached in March (Fig. 2), which corresponded typically to the maximum accumulation of snow on the ground. After a rapid recovery in April, the median gamma ionising capacity fluctuated at around 9.5 cm$^{-3}$ s$^{-1}$ during the rest of the year. The

median radon ionising capacity varied in the range of 0.3-4.1 cm$^{-3}$ s$^{-1}$ and showed a seasonal behaviour very different from that of the gamma radiation. The minimum radon ionising capacity appeared in late spring after a gradual decrease since February. It climbed slowly back in summer and remained at a relatively moderate level of around 1.7 cm$^{-3}$ s$^{-1}$ through autumn till the end of December.





The diurnal cycle in the ionising capacity originated mainly from the radon component and followed variations in the radon activity concentration presented by Chen et al. (2016), which was attributed to the mixing layer development. However, the contribution by gamma radiation shifted the relative levels of the ionising capacity from the seasonal pattern of the radon activity concentration shown by Chen et

al. (2016). A clear diurnal cycle was observed in both spring and summer, with high ionising capacities found in the morning and low ones in the afternoon (Fig. 3). The ionising capacity was generally high in summer and autumn and low in spring and winter, with the median values being 8.9 cm$^{-3}$ s$^{-1}$, 11.2 cm$^{-3}$ s$^{-1}$, 11.3 cm$^{-3}$ s$^{-1}$ and 8.7 cm$^{-3}$ s$^{-1}$ in spring (March-May), summer (June-August), autumn (September-November) and winter (December-February), respectively.

The share of radon ionising capacity in the total ionising capacity was in the range of 10-20% (Fig. 4), with the lowest fraction obtained in spring and a progressive increase through the year reaching the highest share in winter. Interestingly, when separating the data according to the classification of new particle formation (NPF) events defined by Dal Maso et al. (2005), low radon ionising capacities were

found in association with NPF events. The statistical contribution of radon in ion pair production was below 10% on NPF days in all seasons (Fig. 4) and the median radon ionising capacities on NPF event days were typically one third to half of those on non-event days (Fig. S1 and Table 2). This observation is likely related to the fact that marine air masses from Arctic and north Atlantic oceans, which favour NPF (Nilsson et al., 2001), typically have a low radon content (Chen et al., 2016;Wilkening and

Clements, 1975). The mixing layer height (MLH) might affect this phenomenon due to the fact that NPF typically occurs on warm and sunny days with clear skies, whereas non-event days are usually associated with the presence of clouds (Nieminen et al., 2015). Mixing reaches the top of a boundary layer on sunny and clear days, diluting atmospheric radon concentrations and resulting in a lower radon ionising capacity. On cloudy days, mixing is restricted below the clouds, causing a smaller mixing

volume and therefore a higher radon ionising capacity. A statistical summary of the radon and gamma ionising capacities as well as the energy deposited by ionising radiation in the air is presented in Table **3**.





### 3.2. Factors causing variability in the ionising capacity

The seasonal and diurnal variations in the ionising capacity originate from features in ionising radiation, i.e. the atmospheric radon concentration and environmental gamma radiation. Presumably, the

seasonality of the ionising capacity comes from both of the gamma and radon components, whereas the diurnal feature is primarily related to the dynamical response of radon as a result of the mixing layer evolution.

### 3.2.1. Factors affecting the radon ionising capacity

The atmospheric radon concentration, and consequently the derived radon ionising capacity, is affected

by mixing layer dynamics, soil type, soil and meteorological conditions, long-distance transportation, etc. The atmospheric radon concentration is generally related to the mixing layer depth, which is also influenced by varying atmospheric conditions, in terms of the air temperature, wind speed, intensity of solar radiation, etc. These connections are further complicated by the arrival of continental air masses at the measurement site, which brings extra radon in addition to the local sources exhaled from the ground.

Such aspects have been discussed in our earlier work (Chen et al., 2016).

Radon exhalation from the ground typically depends on the availability of radium-226 (Ra-226) – the parent nuclide of Rn-222, the internal structure of mineral grains containing Ra-226, soil type, moisture condition, temperature and ambient pressure (Nazaroff, 1992;Stranden et al., 1984;Lewis et al., 1987;Ashok et al., 2011). Ra-226 typically decays into Rn-222 inside mineral grains. The release of

these bonded radon atoms into pores among soil grains has been found favoured by the presence of some amount of water (Stranden et al., 1984). Nonetheless, a high moisture content in the soil can block the subsequent radon migration through soil pores into the atmosphere (Nazaroff, 1992). The radon's diffusion within and exhalation from the soil have a clear temperature dependency as demonstrated by Chen et al. (2016). A similar relationship was also seen between the radon ionising capacity derived

from the atmospheric radon concentration and the soil temperature: the radon ionising capacity





increased with an increasing soil temperature (Fig. 5). A shallow mixing layer permits a small mixing volume, leading to a high radon ionising capacity, whereas a deep mixing layer promotes dilution. Accordingly, the lowest radon ionising capacities were observed in the highest MLH range. High radon ionising capacities, calculated as medians in 1 °C soil temperature bins, were found at sub-zero soil

temperatures with low MLHs, which are likely related to the reduced clogging by liquid water due to soil freezing and suppressed atmospheric mixing. Liquid water can remain in a frozen soil as thin films adsorbed on soil particles or in crevices and pores well below the freezing point [e.g. (Spaans and Baker, 1996)], which may still encourage the emanation of radon from soil grains. A further exhalation of radon into the atmosphere is probably achieved via frost-induced cracks.

In order to exam the impact of SWC on the radon ionising capacity, a soil temperature window ($T_{soil} >$ 14 °C) was selected when the MLH was restricted in the range of 1000-1500 m, where the influence of soil temperature on the radon ionising capacity could be considered negligible (Fig. 6 a). By zooming into this part of the data, there seemingly existed an effect of SWC, with the highest radon ionising capacity occurring at SWCs of around 0.20-0.25 $m^3$ $m^{-3}$ (Fig. 6 b). This observation is consistent with

the findings by Stranden et al. (1984), who showed that the presence of about 25% of water exerted the biggest enhancement in the exhalation rate of radon from soil samples. Further analysis into the SWC effect was performed by normalising the radon ionising capacity with proxies derived from MLH, soil temperatures and wind direction to minimise the influences of these factors. However, no systemic pattern was isolated between the radon ionising capacity and SWC for a supportive interpretation to be

drawn upon. Besides, although RH has been reported as an influencing factor for radon exhalation (Ashok et al., 2011), no clear correlation between the radon ionising capacity and RH was revealed based on our dataset after shading the variability in the radon ionising capacity brought by MLH, temperature and wind.

### 3.2.2. Factors affecting the gamma ionising capacity

Compared with the radon ionising capacity, the gamma ionising capacity exhibits a simpler pattern. Little diurnal variations exist in the total gamma radiation, and therefore in the derived gamma ionising capacity. However, occasionally high gamma radiation is perceivable during rain events on a temporary





basis, generally of about 2 hours. Such observations are typically ascribed to gamma emissions of the washed-out short-lived progeny of radon (Brunetti et al., 2000;Dwyer et al., 2012;Paatero and Hatakka, 1999). As for the seasonal cycle, the low gamma ionising capacity in winter results from the attenuation effect of snow cover on the terrestrial fraction of gamma radiation. An exponential reduction was

typically seen in the gamma ionising capacity along with snow cover thickening (Fig. 7). A similar feature has been reported on the relationship between snow water equivalents (SWEs) and gamma dose rates for two other Finnish measurement sites in Sodankylä (67°22´N, 26°39´E) and Tikkakoski (62°24´N, 25°40´E) (Hatakka et al., 1998;Paatero et al., 2005). According to Paatero et al. (2005), the constant term in the exponential fitting could represent an approximation of the contribution by cosmic

radiation to the ionising capacity, being about 3 cm$^{-3}$ s$^{-1}$ obtained from fittings to the measured snow depth data. However, 2 cm$^{-3}$ s$^{-1}$ has been generally accepted as the ionisation rate of cosmic radiation for the production of small ions at sea level (Hensen and Hage, 1994), which is only two thirds of the cosmic contribution to the ionising capacity determined with the exponential fittings. This difference might suggest that about one third of the electric charges produced by cosmic radiation in the lower

atmosphere are likely lost in recombination or other sink processes, with the other two thirds survived and descended into small air ions of measurable sizes.

### 3.3. Ionising radiation and observed air ions

Cluster ions are produced from primary and more complex molecular ions via their attachment to pre-existing small neutral clusters and the growth by vapour uptake. Molecular ions include both primary

ions and those originating from the fraction of primary ions that have survived from the recombination or other sinks, and are therefore in a close linkage with the ionising capacity. However, due to technical limitations, no reliable measurement can be carried out to acquire the concentration of molecular ions. Ions in the cluster size range (0.8-1.7 nm) are the smallest detectable air ion group based on the current counting technology. Since the formation of these ions are directly related to the dynamics of molecular

ions and, to certain extent, to the ionising capacity, the focus in this section is leaned onto the analysis of variations in the air ion concentration of the cluster size range in association with the ionising capacity.



### 3.3.1. Patterns in the ionising capacity and cluster ion concentration

The cluster ion concentration exhibited some degree of association with the ionising capacity (Fig. 8). The median cluster ion concentration showed little diurnal variations during the cold months: from January to March, it remained at a low level (Fig. 8 a), with a median of 513 $cm^{-3}$. Clear diurnal cycles were discernible between April and October in cluster ions, while simultaneously the magnitude of diurnal variations in the median ionising capacity became distinguishable. On average, cluster ion concentrations were high during darker hours and low during brighter hours, which were aligned with the general feature seen in the ionising capacity (Figs. 3 and 8b), especially during the booming season of vegetation (May-August).

The enrichment in the median cluster ion concentration in the evening between May and August occurred typically a few hours ahead of the recovery of the median ionising capacity (Fig. 8). Such increases in cluster ion concentrations could result from either an augmentation in the production or a recession in the consumption or sink of these ions. In the former case, a balance between electric charge production and acquisition is re-established towards a higher production of cluster ions. Typically, this can be achieved via either a promotion in the production of electric charges or an enhancement in the charge acquisition. However, since no remarkable increase in the ionising capacity was observed, when cluster ion concentrations started to increase (Fig. 8), the production of cluster ions could be attributed to an enhancement in the charge acquisition. For the formation of cluster ions, the charge acquisition may occur at three stages: a) prior to clustering via the formation of molecular ions from primary ions, b) during the actual clustering process from both primary ions and molecular ions, and c) after clustering via the charge uptake from both primary ions and molecular ions. Recombination consumes part of these acquired charges, leaving the rest retained eventually in the form of cluster ions. In the latter case, there is a hindrance in processes removing ions from the cluster size range, which can be either the growth of cluster ions to bigger sizes than 1.7 nm or the sink of cluster ions to bigger particles. All of these mechanisms, however, are manipulated primarily by atmospheric conditions. Atmospheric conditions, such as the temperature and relative humidity, can directly influence the rate of clustering and growth. Nonetheless, they also modify atmospheric compositions by altering the





availability of precursor gases and the production of functional vapours involved in clustering or growth. As a consequence, these phenomena, seen in Fig. 8, are likely brought by a synergy of complicated atmospheric dynamic processes.

Although high ionising capacities were found in the morning, on average the morning cluster ion
concentration was not so high as the evening level during the relatively warm months between April and October (Fig. 8 a). Especially in autumn months, the enhancement in air ion production from the high morning ionising capacity was not reflected in the cluster ion concentration. Such observations may result from the cluster formation process becoming inferior to the preferred particle growth due to photochemical processes, while facing the dilution led by the expansion of the mixing volume. Autumn
was the second peak period for the occurrence of NPF events at the SMEAR II station after spring (Nieminen et al., 2014). The dissimilar autumn and spring patterns in the cluster ion concentration originate likely from differences in atmospheric conditions and vapour sources. For example, spring UVA radiation intensities are higher than the autumn ones, while the RH shows an opposite behaviour (Lyubovtseva et al., 2005). Biogenic VOC emissions have a strong seasonality (Tarvainen et al.,
2005;Hakola et al., 2012), which is reflected in their atmospheric concentrations. Hakola et al. (2012) demonstrated that monoterpenes tend to dominate VOCs in late summer and autumn, while aromatic hydrocarbons dominate in spring and early summer. Sesquiterpene emissions and concentrations were found to be high in late summer and autumn (Hakola et al., 2012;Tarvainen et al., 2005, 2007).

### 3.3.2. Variations in cluster ion concentrations in sub-size ranges

Ion concentrations in different sub-size ranges (0.8-1 nm, 1-1.2 nm and 1.2-1.7 nm) of cluster sizes showed distinct patterns (Fig. 9). While the positive polarity dominated the overall cluster ion concentration, more negative ions were seen in the first two sub-size ranges (0.8-1 nm and 1-1.2 nm). 0.8-1 nm ion concentrations were found to be the lowest (around 100 $cm^{-3}$) in all seasons. There was a difference of 150-200 $cm^{-3}$ between the concentrations of 0.8-1 nm ions and 1-1.2 nm ions through all
seasons in both polarities, with a larger difference observed in summer and smaller in winter. A more pronounced seasonality was observed for 1.2-1.7 nm ions, with the summer concentrations being the highest and winter concentrations being the lowest.



Slight diurnal patterns were found in 0.8-1 nm and 1-1.2 nm ions. The 0.8-1 nm ion concentration showed features similar to those in the ionising capacity (Figs. 9 and 3), being high in the morning and low in the afternoon, possibly indicating a dominant population of molecular ions in this size range. The seasonal pattern of 1-1.2 nm ions was inconsistent with that of the ionising capacity. Two bumps existed in the concentration of 1-1.2 nm ions in spring and summer, with them found more separated in summer than in spring. However, no identifiable diurnal variation in the 1-1.2 nm ion concentration was seen in autumn; and in winter, the 1-1.2 nm ion concentration showed only a small valley in the late afternoon. For 1.2-1.7 nm, which usually is the size range of critical clusters (Kulmala et al., 2012;Lange et al., 1996;Sipilä et al., 2010;Kulmala et al., 2013), clearer variations could be identified. In all the seasons, a peak close to the sunset, evolving from noon in winter to late evening (21:00) in summer, was observed.

### 3.3.3. Ionising capacities vs. 0.8-1 nm ions

The ionising capacity contains radon and gamma fractions, both of which were observed to have the capability to promote the production of 0.8-1 nm ions (Fig. 10). The radon ionising capacity showed a slightly better correlation with the negative 0.8-1 nm ion concentration than with the positive polarity. Along with the increase in the radon ionising capacity, more of the 0.8-1 nm ions were detected, but the degree of dispersion intensified in the correlation plots (Figs. 10a and S2). This dispersion could come from a joint effect of the temperature ($T$), moisture and background aerosol scavenging. Higher 0.8-1 nm ion concentrations were typically observed during darker hours (Fig. 8 a) when the mixing layer is thin and photochemical processes restricted, possibly because the freshly formed 0.8-1 nm ions are confined in a smaller mixing volume and the production of vapours capable of growing small clusters becomes insufficient to support the growth of these ions out of the size range in question. According to Duplissy et al. (2016) and Kirkby et al. (2011), lower temperatures favour cluster formation in the atmosphere. However, since dark conditions prohibit the further growth of the newly-formed clusters, a build-up of 0.8-1 nm ions during the darker hours is likely enabled at low temperatures, resulting in the highest 0.8-1 nm ion concentrations at the lowest air temperatures in Figs. S2 c and d.





At high radon ionising capacities, the 0.8-1 nm ion concentration, especially in the positive polarity, dropped to a medium level in winter months under moist conditions (Figs. 10 a-d). This observation may be related to the proton affinity of water molecules ($H_2O$), which assigns $H_2O$ the ability to bind

positive charges. The formed cations may constitute a portion of hydronium ions ($H_3O^+$), which are too small to be detected by the BSMA, resulting in the flattening-out of the 0.8-1 nm ion concentration. Also the relatively high CS, as seen in Figs. S2 e and f, could contribute to the lower 0.8-1 nm ion production at high radon ionising capacities. For the negative polarity, a weak positive correlation between the radon ionising capacity and the 0.8-1 nm ion concentration was still perceptible at high

radon ionising capacities, which may result from the ability of $H_2O$ to accommodate negative charges via the formation of clusters, even though it has negative electron affinity (Jalbout and Adamowicz, 2001;Rienstra-Kiracofe et al., 2002). However, in addition to the CS effect, such dispersion could also originate from the variability in the stability of the anion clusters that is influenced, for example, by the geometric configuration of the cluster components as well as dipole properties of them.

The relationship between the gamma ionising capacity and 0.8-1 nm ion production was temperature dependent (Figs. 10 e and f). When the $T$ was below 5 °C, the 0.8-1 nm ion concentration showed a linear relation with the gamma ionising capacity. Similar to the temperature effect on the relationship between the 0.8-1 nm ion concentration and the radon ionising capacity, more 0.8-1 nm ions appeared

also at lower temperatures, corresponding to an unaltered gamma ionising capacity level. A slightly higher correlation coefficient was found between the 0.8-1 nm ion concentration and the gamma ionising capacity in the positive polarity than that in the negative polarity (Figs. 10 e and f). However, when the $T$ was above 5 °C, the linear relationship between the 0.8-1 nm ion concentration and the gamma ionising capacity was no longer identifiable: 0.8-1 nm ion concentrations remained at the lowest

around 70 $cm^{-3}$ in the negative polarity and 30 $cm^{-3}$ in the positive polarity, regardless of the increase in the gamma ionising capacity.





The connections between the overall ionising capacity and the 0.8-1 nm ion concentration are reflected in their diurnal behaviour, with minor dissimilarities associated with varying atmospheric conditions and dynamical processes. The median diurnal variation of the 0.8-1 nm ion concentration was very similar to that of the ionising capacity in spring, summer and autumn: both the lowest median 0.8-1 nm

ion concentration and the ionising capacity occurred typically at 14:00 when the mixing layer was fully developed (Fig. 11). In comparison with the ionising capacity, however, the 0.8-1 nm ion concentration, seemed to follow more instantly the changes in the MLH. The reason behind this observation may be related to the fact that the diurnal variation in the ionising capacity is primarily contributed by radon decay emissions; for example, Chen et al. (2016) showed that the atmospheric radon concentration at

near ground level does not respond immediately to mixing layer expansion or shrinkage. In contrast, 0.8-1 nm ion concentrations are related, in addition to the ionising capacity, also to photochemical processes and availability of nucleating vapours influenced by solar intensity and atmospheric conditions.

In summer and spring, the median 0.8-1 nm ion concentration built up with the increase in the ionising capacity in the early morning before the solar irradiance started to intensify and the mixing layer to grow. The turbulence introduced by mixing layer development may assist the production of vapours for nucleation and growth from photochemical reactions, possibly initiating the growth of these 0.8-1 nm ions. As can be seen from Figs. 9 and S3, peak concentrations in the 1-1.2 nm size range occurred

typically later than those in the 0.8-1 nm size range. Also tiny bumps in 1.2-1.7 nm ion concentrations could be discerned with some time lag at around 8:00-12:00 in spring and 5:00-7:00 in summer.

After the mixing layer had fully developed in spring, summer and autumn, the 0.8-1 nm ion concentration (especially of the positive polarity) showed a slight increase along with the increase in the

ionising capacity and the shrinkage of the mixing layer (Fig. 11). This observation may be attributed to the production of certain vapours that compete with the recombination process and other sink





mechanisms for electric charges either via clustering or simple charge binding. Ionising radiation can potentially free a large number of electric charges, which would 'sacrifice' themselves mostly in recombination, if not otherwise become detectable air ions. The survived electric charges take part in the formation of 0.8-1 nm ions mainly in the form of primary ions and molecular ions. Charge-binding

vapours take over charges from primary ions to form molecular ions. This charge transfer process may also be accompanied by chemical reactions between the vapour molecules and primary ions. Some of the molecular ions are possibly born with a size falling in the 0.8-1 nm size range. Ions in the 0.8-1 nm size range may additionally originate from a) further growth of molecular ions via chemical reactions, b) clustering of primary ions or molecular ions with nucleating vapours or among themselves, or c)

charge uptake by small neutral clusters. Accordingly, the enhanced production of 0.8-1 nm ions after the complete development of the mixing layer, seen in Fig. 11, may be related to changes in the availability of nucleating or charge-binding vapours, altered likely by atmospheric conditions and mixing volume reduction.

Although the ionising capacity continued to increase ever since the recovery in the late afternoon, the enrichment of the 0.8-1 nm ion population ceased typically when little solar radiation was left (Fig. 11). At the same time, bursts in 1-1.2 nm ion concentrations and subsequently in 1.2-1.7 nm ion concentrations were seen (Figs. 9 and S3), which were probably linked to the nocturnal cluster formation events (Lehtipalo et al., 2011;Ehn et al., 2010;Mazon et al., 2016). The emergence of this

phenomenon lies at the basis that the production rate of 0.8-1 nm ions by clustering or charge binding is overtaken by the consumption rate of them via either coagulation or condensational growth. The 1.2-1.7 nm ion concentration typically peaked nearly right after the die-out of photochemical reactions when dark hours came (Fig. S3). Concurrently, the 0.8-1 nm ion concentration started to increase again, as the ionising capacity intensified (Fig. 11).

**3.3.4. Role of ionising radiation in cluster ion formation**

There existed a weak relation between the total ionising capacity and the ion concentration of the whole cluster size (0.8-1.7 nm) range (Fig. 12). On NPF event days, the cluster ion concentration showed a





relatively clear increase with the intensification of the ionising capacity (Figs. 12 a and b). However, since the cluster ions are very small, they can preserve some properties of gaseous molecules and therefore may sink onto bigger particles. Consequently, corresponding to an ionising capacity value, the cluster ion concentration spanned over a wide range, with high cluster ion concentrations occurring at

low CSs. On non-event days, however, the connection between the ionising capacity and the cluster ion concentration became even less identifiable, probably due to the fact that CSs on non-event days are typically higher than those on event days by a factor of 3.6 on average (Dal Maso et al., 2005). In addition to this, meteorological and atmospheric conditions are also more versatile on non-event days than on event days, because NPF events are usually focalised in spring, but non-events are spread all

over the year (Dal Maso et al., 2005;Nieminen et al., 2014).

By carefully setting constraints to focus on data obtained under relatively uniform conditions, it was possible to observe a clear relationship between the total ionising capacity and the whole cluster population (Fig. 13). A time window between 0:00 and 3:00 was selected to minimise the effect of

diurnal variations. Since the value of CS is typically largest during night-time (Kulmala et al., 2013), the constraint on the CS was set to be below 0.002 s$^{-1}$, instead of 0.001 s$^{-1}$ in Fig. 13. No dependency of the cluster ion concentration on the CS or on the hour of the day was identified. As can be seen in Fig. 2, generally little variations in both radon and gamma ionising capacities existed between September and December, which therefore was chosen as the time window for the analysis. Furthermore, the wind

direction was restricted in between 280° and 30° to shield the effect of transported radon sources from continental areas.

On both NPF event (Figs. 13 a and b) and non-event (Figs. 13 c and d) days, the cluster ion concentration grew with an increase in the ionising capacity. The correlation between the ionising capacity and the cluster ion concentration was weaker on non-event days than that on NPF event days.

Higher ionising capacities were seen on non-event days, and correspondingly more cluster ions were detected on such days than on event days. The cluster ion concentration tended to level up on event days at high ionising capacities for both polarities ($\geq$10.5 cm$^{-3}$ s$^{-1}$) (Figs. 13 a and b), implying that the





formation process of cluster ions was not efficient enough to rescue charges from recombination. However, such a feature showed no trace on non-event days and also the cluster ion concentration was more dispersed as a function of the ionising capacity. The reason for these observations may be owing to the fact that new particle formation occurs on days with some certain combination of atmospheric

conditions as well as physical and chemical processes, but high variability is involved in these parameters on non-event days.

## 4. Conclusions

The ionising capacity was defined as the potential production rate of ion pairs in the atmosphere by the ionising radiation. We presented seasonal and diurnal patterns in the ionising capacity and air ion

concentration in the lower atmosphere, based on ambient measurement data collected during 2003-2006 from a boreal forest site in southern Finland. Factors responsible for variations in the ionising capacity were overviewed and we demonstrated clearly the promotion effect of the ionising capacity on air ion production via in-depth analysis into their connections. In principle, the production of cluster-size (0.8-1.7 nm) air ions is more closely associated with molecular ions than with the ionising capacity.

However, modern ion spectrometers are not yet capable of providing the number size information on air ions smaller than 0.8 nm in mobility diameters. Thus, no direct evidence bridging the variability in the cluster ion concentration and ionising radiation could yet be acquired. Nevertheless, air ions detected in the lowest size band (0.8-1 nm) of the cluster size range (0.8-1.7 nm) exhibited interesting connections to variations in the ionising capacity. The evolvement of these 0.8-1 nm ions with time to larger sizes in

the cluster size band was also identified, affirming the role of ionising radiation in the production of air ions in the lower atmosphere. Yet still, the number size distribution of air ions smaller than 0.8 nm is a necessity to further obtain insights into air ion formation mechanisms from ionising radiation. For this purpose, theoretical understanding on the formation mechanisms of cluster ions from molecular ions should be deepened. In addition, also advancing instrumental development for the detection of sub-0.8

nm ions should be brought onto the agenda.



## 5. Acknowledgement

This work received funding supports from the Academy of Finland Centre of Excellence (project no. 272041 and 1118615), European Union's Horizon 2020 research and innovation programme under grant agreement No. 654109 (ACTRIS-2) as well as the European Union Seventh Framework

Programme (FP7/2007-2013 ACTRIS) under grant agreement No. 262254. Also the CRyosphere-Atmosphere Interactions in a Changing arctic Climate (CRAICC) project of the Nordic Centre of Excellence is acknowledged. The authors appreciate the valuable communication with Prof. Jaana Bäck and Dr. Pasi Kolari.

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





**Tables**

Table 1. Decay modes and energy of radon-222 and its short-lived progeny taken into account in the ionising capacity determination. Decay partitioning is accounted in defining the weighted average decay energy. The data are extracted from the National Nuclear Data Centre of Brookhaven National
5  Laboratory (http://www.nndc.bnl.gov/chart/).

| Nuclide | Decay mode | Weighted average decay energy [keV] |
|---|---|---|
| $^{222}$Rn | α (100%) | 5589 |
| $^{218}$Po | α (99.98%) | 6001 |
| $^{214}$Pb | β$^-$ (100%) | 225 |
| $^{214}$Bi | β$^-$ (100%) | 642 |
| $^{214}$Po | α (100%) | 7687 |

Table 2. Median radon ionising capacity in cm$^{-3}$ s$^{-1}$ on new particle formation (NPF) event and non-event days in different seasons over 2003-2006.

| NPF classification | Event | Non-event |
|---|---|---|
| Spring | 0.68 | 1.31 |
| Summer | 0.90 | 1.74 |
| Autumn | 0.99 | 1.8 |
| Winter | 0.56 | 1.96 |





Table 3. Energy in eV m$^{-3}$ s$^{-1}$ deposited in the air by total environmental gamma radiation ($E_\gamma$) and by alpha and beta activities from radon-222 decay ($E_{Rn}$) based on the 2003-2006 data. The gamma and radon ionising capacities ($Q_\gamma$ and $Q_{Rn}$) in cm$^{-3}$ s$^{-1}$ were derived from the deposited energy assuming 34 eV for the generation of one ion pair in the air.

|  | Min | 25% | 50% | 75% | Max | Mean | STD |
|---|---|---|---|---|---|---|---|
| $E_\gamma$ | 0.769 | 242 | 314 | 338 | 544 | 295 | 72.3 |
| $E_{Rn}$ (α&β) | 0.145 | 28.6 | 50.3 | 79.5 | 284 | 58.1 | 38.8 |
| $Q_\gamma$ | 4.34 | 6.70 | 9.09 | 9.34 | 11.6 | 8.14 | 1.80 |
| $Q_{Rn}$ | 0.00427 | 0.84 | 1.48 | 2.34 | 8.35 | 1.71 | 1.14 |





**Figures**

Figure 1: primary ions, molecular ions and cluster ions.

Figure 2. Seasonal patterns of radon and gamma ionising capacities as a function of day-of-year over the years 2003-2006. The radon ionising capacity was determined from the alpha and beta radioactivity associated with radon-222 decay and the gamma ionising capacity from total gamma radiation. The data are presented as daily medians.

Figure 3. Diurnal cycles of the total ionising capacity presented as medians in different seasons in the years 2003-2006. Spring: March-May, summer: June-August, autumn: September-November, and winter: December-February.

Figure 4. The relative importance of alpha and beta activities from radon-222 and its short-lived daughter nuclides as well as total gamma radiation in air ion production. The division of the share between radon and gamma fractions on all days is indicated by a plain line, on new particle formation (NPF) event days by a line with an open circle and on non-event days by a line with a cross.

Figure 5. The median radon ionising capacity as a function of the soil temperature in 1 °C bins. Lines are color-coded by mixing layer height (MLH) intervals.

Figure 6. The effect of soil conditions on the radon ionising capacity. a) Radon ionising capacities vs. soil temperatures in the MLH range of 1000-1500 m with soil water content (SWC) shown on the colour scale. Hourly data are presented for 2003-2006 with data in 2004 excluded. According to Ilvesniemi et al. (2010), the quality of SWC data in 2004 was not acceptable. b) Radon ionising capacities vs. SWCs for soil temperatures larger than 14 °C.

Figure 7. The attenuation effect of total gamma radiation by snow cover for the years 2003-2006. Pit 70 and 100 are two measurement points of snow cover depth. Exponential fittings were made with the goodness of fit denoted as $R^2$. The 95% confidence bounds for the constant term are [2.2 4.1] and [1.3 4.0] for pit 70 and pit 100, respectively.





Figure 8. Median variations in a) the mean cluster (0.8-1.7 nm) ion concentrations of negative and positive polarities measured by a balanced scanning mobility analyser (BSMA) and b) the total ionising capacity over the years 2003-2006. Hourly medians were calculated for the whole measurement period from the 10-minute measurement data prior to the processing of median values for each month as a
vector of hour-of-day.

Figure 9. Median seasonality of the cluster ion concentration in 0.8-1 nm, 1-1.2 nm and 1.2-1.7 nm ranges over the years 2003-2006. Negative ion concentrations are depicted by the solid line and positive ion concentrations by the dash line.

Figure 10. Relationship between the hourly ionising capacity and the 0.8-1 nm ion concentration when
the condensation sink (CS) is below 0.001 s$^{-1}$. Upper panel (a and b): radon ionising capacities vs. 0.8-1 nm ion concentrations for Jan. - Oct. (circle) and for Nov. - Dec. (cross). Middle panel (c and d): radon ionising capacities vs. 0.8-1 nm ion concentrations with absolute humidity on the colour scale for Nov. - Dec. Lower panel: gamma ionising capacities vs. 0.8-1 nm ion concentrations with air temperature ($T$) on the colour scale. The correlation coefficients ($R$) in the upper panel were determined
using all available data and in the lower panel were determined between the gamma ionising capacity and the 0.8-1 nm ion concentration, when the $T$ below 5 °C.

Figure 11. Diurnal patterns in median 0.8-1 nm negative and positive ion concentrations, ionising capacities, global and UVB radiation intensities as well as modelled mixing layer heights (MLH) in different seasons over 2003-2006.

Figure 12. The 1-h cluster (0.8-1.7 nm) ion concentration as a function of the total ionising capacity on new particle formation event days (upper panel) and non-event days (lower panel), with the condensation sink (CS) indicated on the colour scale.

Figure 13. The cluster (0.8-1.7 nm) ion concentration as a function of the ionising capacity (radon ionising capacity + gamma ionising capacity) for selected data in the years 2003-2006, with correlation
coefficient (R). The data was constrained on a) & b) event days and c) & d) non-event days in





September-December between 0:00 and 3:00 with the wind direction between 280° and 30°, while the condensation sink (CS) was below 0.002 s$^{-1}$. No dependence of the cluster ion concentration on the CS or hour-of-day was identified. Also the snow season was screened out. The negative polarity is shown in a) and c) and the positive in b) and d), with soil temperatures in the organic layer (-4-0 cm above the mineral soil) shown on the colour scale.

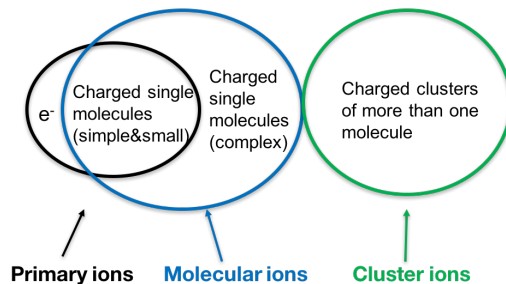

Figure 1: primary ions, molecular ions and cluster ions.





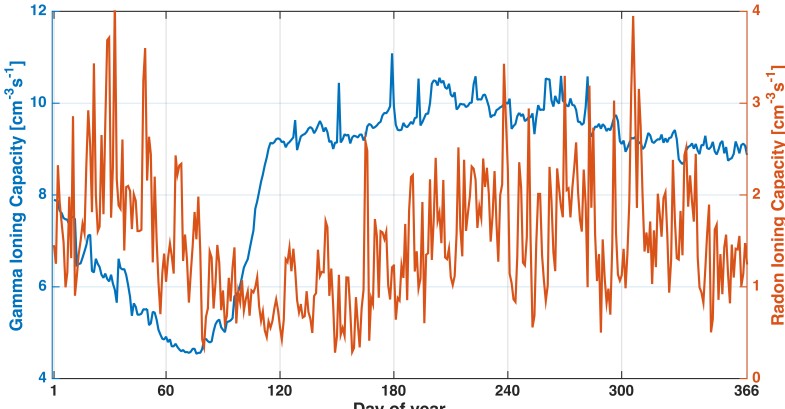

Figure 2. Seasonal patterns of radon and gamma ionising capacities as a function of day-of-year over the years 2003-2006. The radon ionising capacity was determined from the alpha and beta radioactivity associated with radon-222 decay and the gamma ionising capacity from total gamma radiation. The data are presented as daily medians.





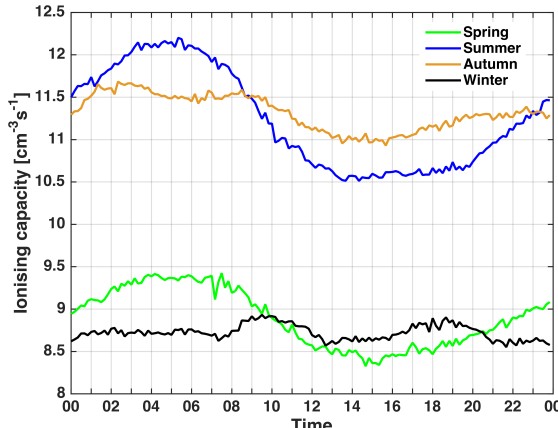

Figure 3. Diurnal cycles of the total ionising capacity presented as medians in different seasons in the years 2003-2006. Spring: March-May, summer: June-August, autumn: September-November, and winter: December-February.





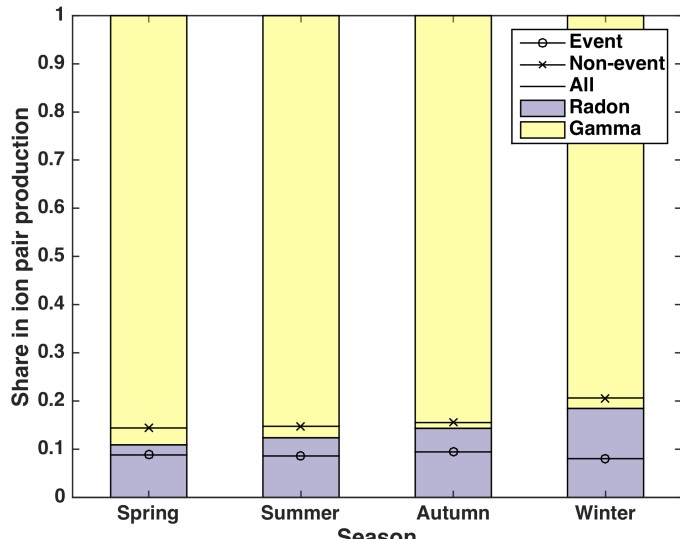

Figure 4. The relative importance of alpha and beta activities from radon-222 and its short-lived daughter nuclides as well as total gamma radiation in air ion production. The division of the share between radon and gamma fractions on all days is indicated by a plain line, on new particle formation
5  (NPF) event days by a line with an open circle and on non-event days by a line with a cross.




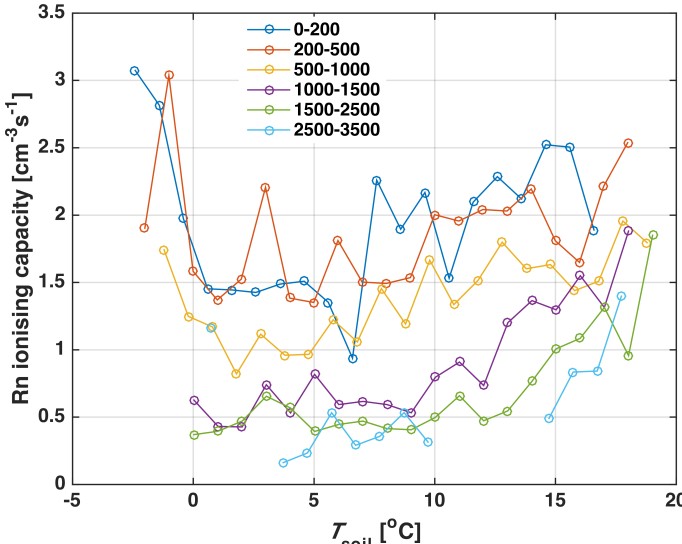

Figure 5. The median radon ionising capacity as a function of the soil temperature in 1 °C bins. Lines are color-coded by mixing layer height (MLH) intervals.





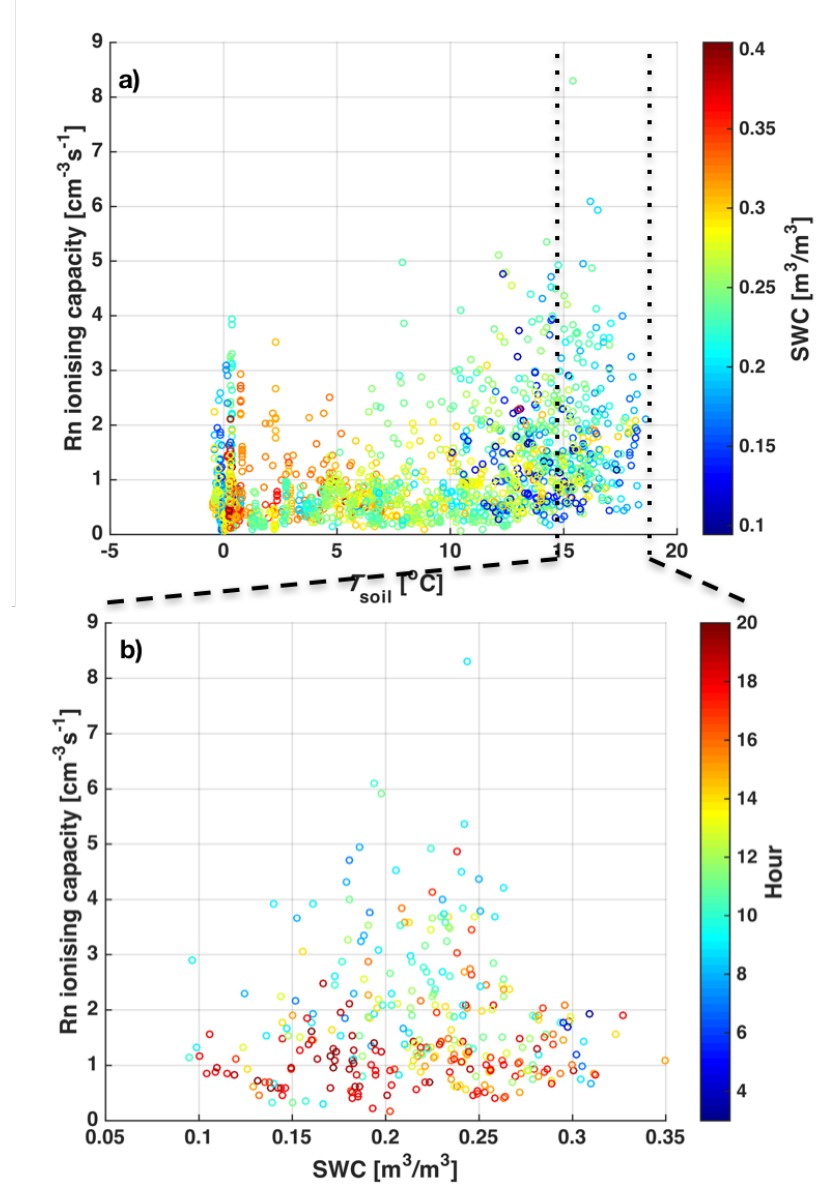





Figure 6. The effect of soil conditions on the radon ionising capacity. a) Radon ionising capacities vs. soil temperatures in the MLH range of 1000-1500 m with soil water content (SWC) shown on the colour scale. Hourly data are presented for 2003-2006 with data in 2004 excluded. According to Ilvesniemi et al. (2010), the quality of SWC data in 2004 was not acceptable. b) Radon ionising capacities vs. SWCs for soil temperatures larger than 14 °C.

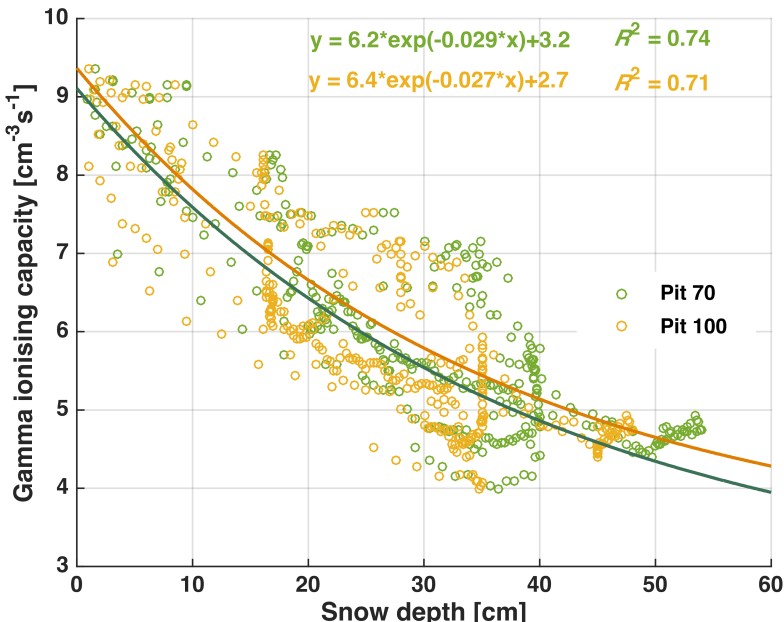

Figure 7. The attenuation effect of total gamma radiation by snow cover for the years 2003-2006. Pit 70 and 100 are two measurement points of snow cover depth. Exponential fittings were made with the goodness of fit denoted as R2. The 95% confidence bounds for the constant term are [2.2 4.1] and [1.3 4.0] for pit 70 and pit 100, respectively.





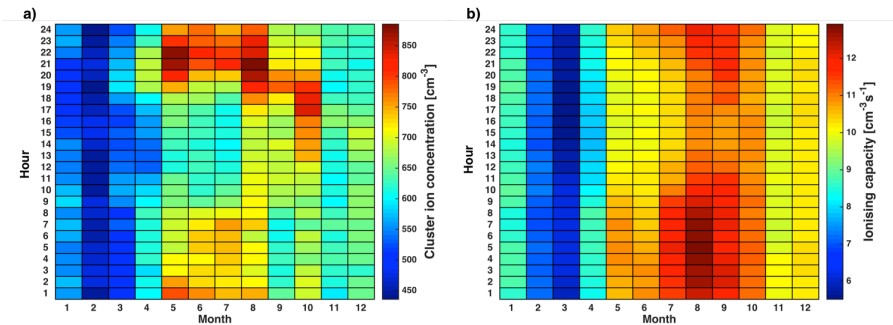

Figure 8. Median variations in a) the mean cluster (0.8-1.7 nm) ion concentrations of negative and positive polarities measured by a balanced scanning mobility analyser (BSMA) and b) the total ionising capacity over the years 2003-2006. Hourly medians were calculated for the whole measurement period from the 10-minute measurement data prior to the processing of median values for each month as a vector of hour-of-day.





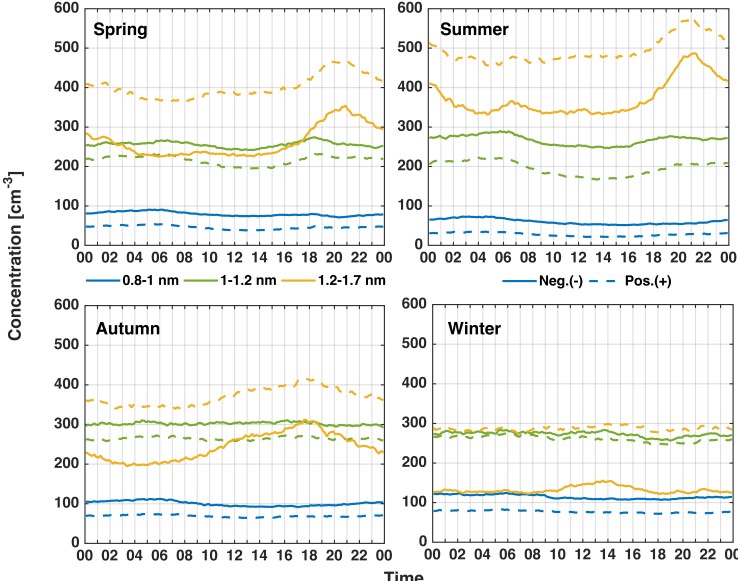

Figure 9. Median seasonality of the cluster ion concentration in 0.8-1 nm, 1-1.2 nm and 1.2-1.7 nm ranges over the years 2003-2006. Negative ion concentrations are depicted by the solid line and positive ion concentrations by the dashed line.




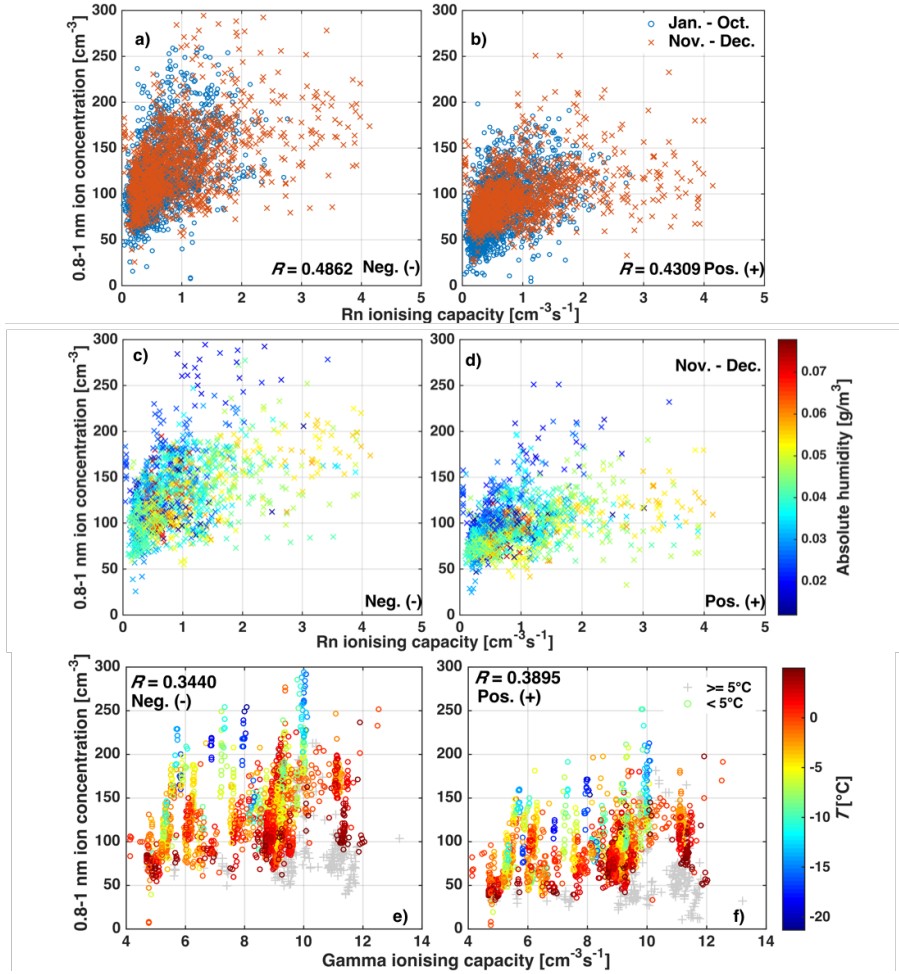

Figure 10. Relationship between the hourly ionising capacity and the 0.8-1 nm ion concentration when the condensation sink (CS) is below 0.001 s$^{-1}$. Upper panel (a and b): radon ionising capacities vs. 0.8-1 nm ion concentrations for Jan. – Oct. (circle) and for Nov. – Dec. (cross). Middle panel (c and d): radon ionising capacities vs. 0.8-1 nm ion concentrations with absolute humidity on the colour scale for Nov. – Dec. Lower panel: gamma ionising capacities vs. 0.8-1 nm ion concentrations with air

temperature ($T$) on the colour scale. The correlation coefficients ($R$) in the upper panel were determined using all available data and in the lower panel were determined between the gamma ionising capacity and the 0.8-1 nm ion concentration, when the $T$ below 5 °C.

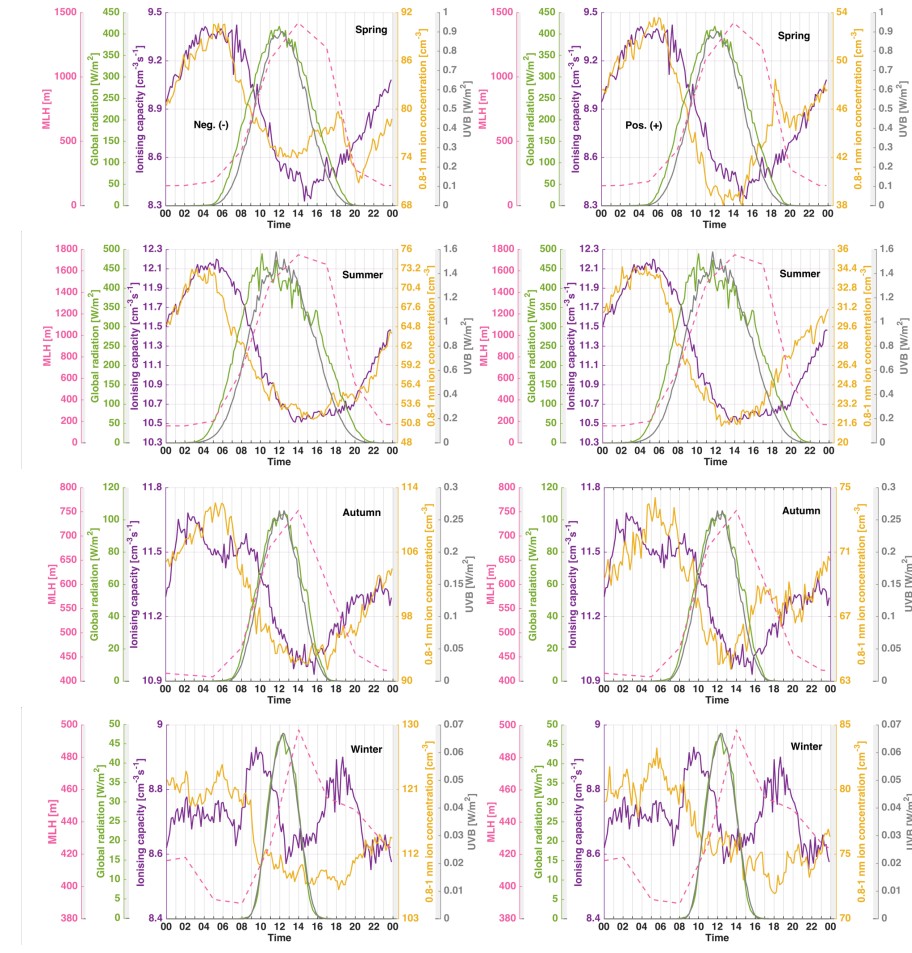

Figure 11. Diurnal patterns in median 0.8-1 nm negative and positive ion concentrations, ionising capacities, global and UVB radiation intensities as well as modelled mixing layer heights (MLH) in different seasons over 2003-2006.





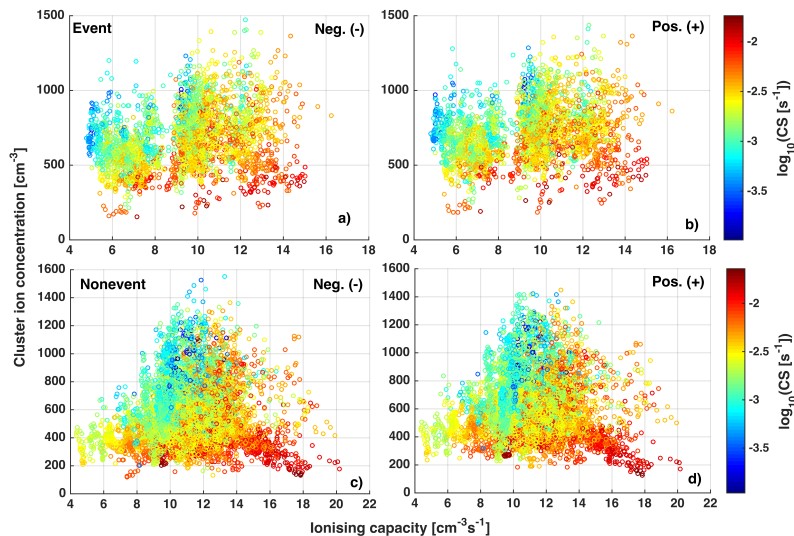

Figure 12. The 1-h cluster (0.8-1.7 nm) ion concentration as a function of the total ionising capacity on new particle formation event days (upper panel) and nonevent days (lower panel), with the condensation sink (CS) indicated on the colour scale.





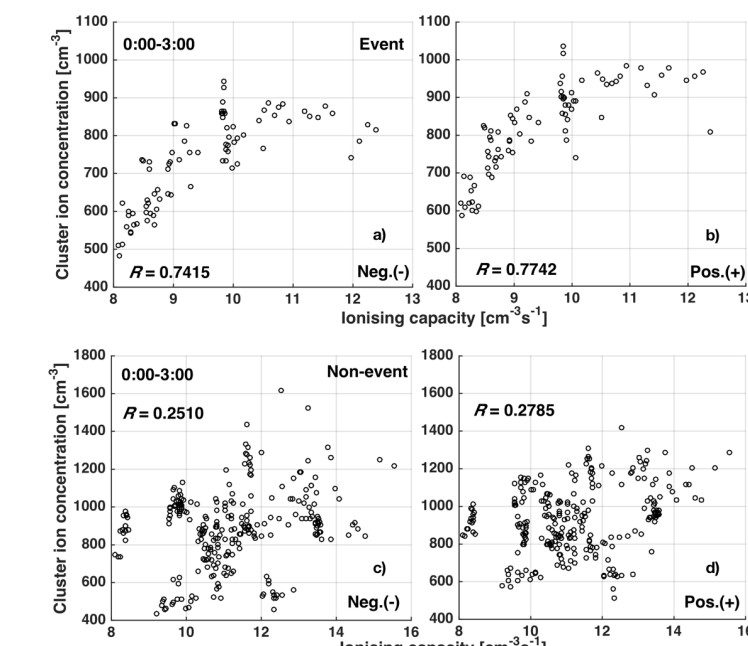

Figure 13. The cluster (0.8-1.7 nm) ion concentration as a function of the ionising capacity (radon ionising capacity + gamma ionising capacity) for selected data in the years 2003-2006, with correlation coefficient ($R$). The data was constrained on a) & b) event days and c) & d) non-event days in September-December between 0:00 and 3:00 with the wind direction between 280° and 30°, while the condensation sink (CS) was below 0.002 s$^{-1}$. No dependence of the cluster ion concentration on the CS or hour-of-day was identified. Also the snow season was screened out. The negative polarity is shown in a) and c) and the positive in b) and d), with soil temperatures in the organic layer (-4-0 cm above the mineral soil) shown on the colour scale.

