# Peer review of "How do air ions reflect variations in ionising radiation in the lower"

_Atmospheric Chemistry and Physics, 2016_

## Referee Comment (RC1) · Anonymous Referee #1 · 28 Jul 2016

In this paper Chen et al present a comprehensive set of atmospheric ion measurements in combination with detailed measurements of radioactivity. The heavily instrumented site used permits a very detailed analysis of the effects of ionising radiation on the concentration and mobility (size) of atmospheric ions, and how the ions are modulated by seasonal and meteorological factors. This is the most thorough analysis of its type that I have seen, and potentially makes a valuable contribution to understanding the role of ionising radiation in the variability of atmospheric ions. Some interesting differences in the contribution of ionising radiation on days when there are new particle events are also identified.

Despite the high quality of the data, there are several deficiencies in the way the paper is written, in particular, an apparent unfamiliarity with the literature in this area. The introductory material is narrow, and implies that (a) this is a topic which has hardly

been worked on and (b) there is limited understanding of how radioactivity makes ions. Since radioactivity and cosmic rays were discovered through their ionisation of the atmosphere at least a hundred years ago, these implications are incorrect. The authors should be referring to some historical material (e.g. the Irish work of Nolan and collaborators) or, at a minimum, a historical review, and the work on ion measurements from other groups in the late twentieth century, to explain how their study progresses the research area. The motivation of the paper needs to be refocused towards the work that is described, which is a far more sophisticated study of the interaction between radioactivity and the properties of atmospheric ions than the introduction suggests. The title of the paper also implies a much more basic study than is actually carried out and might be better changed to something like, "Effects of ionising radiation on ion mobility in a range of atmospheric conditions" (this is just a suggestion and should not be taken literally, but I hope you understand my point).

A second aspect of the paper that indicates lack of awareness of other studies is that the authors present their own definitions for terms that are already precisely defined. For example, they introduce a term called "ionising capacity" which to all intents and purposes appears to be identical to what the rest of the community already calls "ionisation rate", since it has the same units and even the same symbol. The only justification I can think of for bringing in this new terminology would be if the ionising capacity were a theoretical maximum amount of ionisation, given the energies and activities of the particles involved, which may not be the same as the actual ionisation rate. However, since the authors make no attempt to justify their new definition, and also, the only way to measure the "true" ionisation rate would be through detailed ion measurements of the sort made by the authors, making the distinction between "capacity" and "rate" doesn't seem particularly helpful. Some of the energy of the radioactive particles will be used for excitation and not ionisation, but I would imagine that this is a relatively small fraction, and it is not unreasonable to assume that all the energy lost by the radioactive particles goes into ionisation. I recommend that the authors remove the references to and definition of "capacity" and simply talk about ionisation rate for con-
sistency with other work. They could add a caveat mentioning energy loss to excitation if this concerns them.

The wheel is reinvented again in figure 1. Many years of work in the early twentieth century went into defining cluster ions, small ions, large ions, etc. If the authors are going to come up with their own definitions they must explain why they are needed. In particular, the distinction between primary and molecular ions seems arbitrary, since the primary ions N2+ and O2+ are technically molecular ions anyway, and the ions you call molecular ions I thought were unstable in the atmosphere and cluster immediately. Fourthly, the authors do not seem aware of the known theoretical relationships between ionisation rate and ion concentration. The paper they cite by Harrison and Carslaw (2003) contains a good introduction to this theory. The lack of theoretical awareness is particularly apparent when looking at figure 13, which seems to be a very nice demonstration of the ion balance equation in the recombination limit. I've seen similar plots before (e.g. in Aplin and Harrison, Rev Sci Instrum 2000), but this is by far the best data, however the authors do not place this work in context. They need to talk about the ion balance equation and compare their work to those of other groups; this will demonstrate the strength of their data.

The paper presents a huge amount of data is presented, in a long main text, but the conclusions are brief bordering on the obvious. The conclusions need to present a synthesis of the data and put it into context in the conclusions. It is also not clear why the results are important – for example the last sentence of the entire paper recommends that instruments that can measure sub-0.8 nm ions are needed – why? The results on new particle formation are novel and very interesting, but can you discuss their implications?

A final scientific point is that the authors do not adequately explain at the beginning of the paper that they are omitting to measure most of the cosmic rays and that they usually contribute about 20% of the ionisation rate. They are mentioned in the context of high-energy gamma radiation, but there are muons and electrons too which are not
mentioned. Later in the text, it is implied from the residuals that the cosmic ionisation rate is 3 /cc/s, and it is argued that because the "textbook" rate is 2 /cc/s, then one third of the charge created is lost before even becoming atmospheric ions. This seems totally speculative; surely it is much more likely that the fit/measurement errors and the high latitude of the Finnish measurement site and perhaps solar activity could account for the discrepancy. It would be possible to carry out a more quantitative analysis here.

Typographical/minor errors

P3 L12 "generated electric charges from . . . by the derivatives of them" does not make sense

P4 L7-21 Please revise this paragraph in line with the comments above

P5 L15 "descripted" -> "described"

P6 L17 All ion spectrometers are defined by their upper and lower mobility limits, so this sentence is meaningless. Why is the upper mobility limit 3.2 cm-2V-1s-1?

P6 L20 What are the potential consequences of the choice of mobility-diameter relationship and its parameters?

P7 L1-2 please be consistent with units.

P7 L2 how and why is only one eighth of the air sampled?

P7 L13 "programme" -> "program"

P8 L8 Please explain what the condensation sink is and its units; not everyone will be familiar with it.

P8 L6-18 this is quite a lot of meteorological detail for work which is presumably published elsewhere. Could you shorten it and cite a reference?

P8 L22 Notwithstanding the comments above, should these units include a "per unit volume"?

**ACPD**
P9 L15 This statement is meaningless since you derive the "ionising capacity" directly from dose rate in equation 1. Please remove

P10 L19 Can you explain why marine air masses will have a low radon content; again, not all readers will necessarily know why.

P12 L10 "exam" -> "examine"

P12 section 3.2.2 Since radon emits gammas, which are included in your gamma radiation as you said earlier, can you state more clearly that you only mean alphas and betas from radon here?

P14 L8 would be better to say "growth season"

P14 L23 not sure what is meant by, "there is a hindrance"

P14 L24 what is the significance of the 1.7 nm threshold?

P15 L14-18 As it stands, this is irrelevant to the rest of the paper and should be deleted.

P15 L20-27 It is well known that negative atmospheric ions are smaller than positive ones, though this work offers more detail. Please cite the classical literature here.

P16 L16 not sure what is meant by molecular ions here.

P17 L4-6 This statement is speculative and needs to be qualified as such.

P17 L7 You need to add some words on CS here.

P17 L20 what do you mean by "unaltered gamma ionising capacity level"?

P19 L1 what do you mean by "clustering or simple charge binding?"

P19 L20 "focalised" -> "localised"

Table 3: The labels are not adequately explained – could perhaps talk about quantiles to make it clearer what you mean. Please also define STD. As discussed above, it would be better to talk about ionisation rates.

**ACPD**
Figure 2: It would be helpful to provide some indication of the variability, for example by a shaded band around the median.

Figure 6: I don't understand where soil temperatures fit in here, is this a typo?

Figure 7: R2 needs a superscript on the 2

Figure 11: This plot tries to convey too much information. Would it be better to present all the data together (e.g. as a median), or pick one representative year and add the rest to the supplementary information?

Figure 13: A linear fit may not be appropriate here given the theory – please revise

---

## Referee Comment (RC2) · Anonymous Referee #2 · 15 Aug 2016

The paper describes the acquisition and analysis of a long-term (3 year) data set from a boreal forest site on ionising radiation and cluster ions, attempting to ascertain relationships between them and determine factors responsible for their variation in diurnal and seasonal cycles. Both the experimental data and analysis are very extensive and provide useful insights into the role of ionising radiation from radon and progeny, and gamma radiation from the ground, in producing new cluster ions under varying meteorological and seasonal conditions, with a particular focus on new particle formation events. The work presents a significant advance in the understanding of the influence of ionising radiation on cluster ion concentrations, and infers information about the formation process itself, and hence would be a welcome and worthwhile addition to the literature in this research area. However, I feel that some minor revisions are necessary before the paper can be published. Some potential confounding factors e.g. Rn-220

(thoron) and particularly cosmic ray ionisation, data for which are not presented, are not discussed sufficiently and would benefit from being considered in the context of, and/or justified in terms of their exclusion from, the overall analysis. Also, citation of, and comparison of results with, previous work could be improved upon in the Introduction section and in discussion.

**Specific comments**

The paper in general would benefit from additional discussion of previous work, in comparison with results presented, but is required in particular at specific points in the manuscript:

a) In the Introduction, in particular regarding air ion mobility spectrometry, there is little citation of work undertaken outside the groups of the authors and Tammet and co-workers. Substantial though this body of work is, there are other noteworthy contributions to the literature, especially in a historical context, and I encourage the authors to expand upon this in their Introduction.

b) Section 3.3.2 describes "Variations in cluster ion concentrations in sub-size ranges" which, to a certain extent, results in a discussion in part about the ion mobility, since if ion mobility is different (either in response to atmospheric changes e.g humidity, or as a result of different polarity) then ions may be classified into different categories here. There is a reasonable body of work in the literature describing ion mobility and humidity effects which may have a bearing on results presented here, and general conclusions (positive ion concentration > negative concentration, positive ion mobility

spectrometer used (100-3000 keV) and would miss a substantial 'cosmic ray ionisation capacity'. If data on cosmic ray flux at the measurement site is available, this would be a very worthwhile inclusion to the paper, even if it is only discussed in a qualitative manner. For example, were there cosmic ray events or solar energetic events during the measurement period which might have influenced ionisation in a manner not accounted for in the Rn and gamma ionisation capacity data? What proportion of overall ionisation might be caused by sources other than the Rn and gamma ionisation captured in the data presented, and is this expected to be constant throughout the study or variable? This discussion might be particularly relevant given the high latitude of the measurement site.

Also, it may well be that there is not sufficient ionisation by thoron decay to justify including this as a factor in the study, and indeed this has been suggested in previous work reported from Hyytiälä (Laakso et al. (2004) ACP 4, 1933) but the possibility should at least be mentioned here, even if only to clarify it is insignificant.

I don't think Figure 1 in its current form is necessary. The concepts are adequately described in the text and the figure does not add much value to the description. If an 'overview' figure is deemed helpful by the authors, perhaps a schematic introducing the processes involved in cluster ion formation and growth due to ionisation, vapour condensation, recombination etc. would be more instructive.

Section 3.1 and Figures 2-4 show seasonal analysis, splitting the annual cycle into 4 3-month periods. However, looking at Figure 2, the 'Spring' period shows a significant change in, in particular, the gamma ionising capacity. So, for example while the average ionising capacity (gamma + Rn) is similar to Winter when taken across this period March-May as a whole (Figure 3), the ionising capacity increases significantly during this period and also the relative importance of gamma and Rn changes. It may be worth repeating some of the analyses only for, say, mid-February to end-March, to examine this specific period where Rn contributes the largest fraction to overall ionising capacity across the whole year, as I wonder whether features are being diluted and missed by
taking the average over a period in which large changes in these parameters occur.

P4 L25 Please clarify whether 'during 2003-2006' means whole-year data collection for these years, or if not, in which months data collection/use began/ended.

P6 I note also that APi-TOF may not provide direct information on the actual electrical mobility of atmospheric cluster ions, as encountered in environmental measurements using air ion spectrometers, because of the evaporation of clustering water molecules during the APi-TOF measurement.

P7 Please state the voltage scan time for the BSMA in this work (it looks like it appears in the legend to Figure 8 but should be specified in the main text as well).

P8 L11 Please clarify your justification for using only 'O horizon' soil data.

Minor technical corrections

P4 L25 "Monitoring devices for" not "The monitoring devices of the"

P5 L15 "described" not "descripted"

P8 L8 "Time-Domain Reflectometer" or "...Reflectometry (TDR) device."

P12 L10 "examine" not "exam"

P12 L22 what is meant by "shading the variability"?

P13 L9-11 move text "to the measured snow depth data" to L9: "exponential fitting ... could represent" or otherwise reword, to clarify what is relationship is being probed.

P14 L9 "blooming" not "booming"?

P20 L9 "localised" not "focalised"?

P20 L26 "level out" not "level up"

P22 L2 "support" not "supports"

ACPD
P22 L19-22 the full citation now appears to be available online, please amend.

P33 L4 "0-4 cm" not "-4-0 cm"

P34 Fig 2, both y-axes "ionising" not "ioning"

P43 L3 "T was below" not "T below"

---

## Author Comment (AC2)

We would like to express our appreciation to referee #2 for the careful review of our work. The referee's comments are ordered by number in this document. For each critical point presented by the referee, our reply consists of a general response and an indication of changes in the manuscript following the guidelines.

1. Some potential confounding factors e.g. Rn-220 (thoron) and particularly cosmic ray ionisation, data for which are not presented, are not discussed sufficiently and would benefit from being considered in the context of, and/or justified in terms of their exclusion from, the overall analysis.

   **The reviewer is right. We added information concerning these aspects. More detailed description of these changes can be found under points 4 and 5.**

2. In the Introduction, in particular regarding air ion mobility spectrometry, there is little citation of work undertaken outside the groups of the authors and Tammet and co-workers. Substantial though this body of work is, there are other noteworthy contributions to the literature, especially in a historical context, and I encourage the authors to expand upon this in their Introduction.

   **Thank you for the comment! We added description of other instrumentations for air ion study in the revised manuscript as follows**

   **'Devices employed for ion studies comprise different types of aspiration condensers, ion mobility spectrometers (IMS) and mass spectrometers (Hirsikko et al., 2011;Tammet, 1970;Laskin et al., 2012;Cumeras et al., 2015). Notably, modern key instruments for field observations of air ions are mainly aspiration contender-based devices and mass spectrometers, such as the Gerdien counter - an integral aspiration condenser (Gerdien, 1905;Aplin and Harrison, 2000;Vojtek et al., 2006), ion spectrometers designed by Airel Ltd – single or multiple channel aspiration condensers (Tammet, 2011, 2006;Mirme et al., 2007;Manninen et al., 2009;Kulmala et al., 2016) and the atmospheric pressure interface time-of-flight mass spectrometer (APi-ToF) (Junninen et al., 2010). While aspiration condensers provide information on the concentration and mobility of charge carriers, mass spectrometers reveal mainly the chemical properties of them. The IMS, however, has a limited application in studying ambient ions, due to difficulties in spectrum interpretation (Hirsikko et al., 2011). The purpose of these instrumentations is not only limited to air ion or air conductivity observations, but also for nano-material synthesis**

(Kruis et al., 1998) and for the improvement of the fundamental understanding on the relationship between mobility, mass and size (Ku and de la Mora, 2009).
'

3. Section 3.3.2 describes "Variations in cluster ion concentrations in sub-size ranges" which, to a certain extent, results in a discussion in part about the ion mobility, since if ion mobility is different (either in response to atmospheric changes e.g humidity, or as a result of different polarity) then ions may be classified into different categories here. There is a reasonable body of work in the literature describing ion mobility and humidity effects which may have a bearing on results presented here, and general conclusions (positive ion concentration > negative concentration, positive ion mobility < negative mobility) are in common with several previous results.

**Thank you for the comment! Ion mobility can indeed be influenced by ambient humidity and there are many articles written on this issue. The humidity effect is a very critical issue, for example, in interpreting spectra from ion mobility spectrometers. However, we dealt with data collected from an aspiration condenser-type instrument and the device measured the number size distribution of ambient ions. Under different humidity (Canton, 1753)conditions, an ion, e.g. a H2SO4-H2O cluster, can be classified into a different size category due to the formation of a new cluster between the original H2SO4-H2O and water molecules, known as the humidity influence. However, this is a reflection of the modification of the ambient size distribution by moisture. The data itself is still representative of the atmospheric distribution of air ions.**

**We could observe some connection of the 0.8-1 nm ion concentration to the absolute humidity, with a negative proportionality. However, there was no reversed proportionality found in the adjacent size range (1-1.2 nm) in relation to the absolute humidity. This might be attributed to the fact that atmospheric clusters have too much variability in their structures and compositions and also complications from atmospheric conditions may further hinder the humidity effect from being seen.**

**Some comparisons with previous results reported by other researchers are included in the revised manuscript.**

'While the positive polarity dominated the overall cluster ion concentration, more

negative ions were seen in the first two sub-size ranges (0.8-1 nm and 1-1.2 nm). The former results from the electrode effect of the negatively charged earth surface, which repels negative ions in its vicinity; and it is a well-known phenomenon to the atmospheric electricity community (Israël, 1970;Wilson, 1921;Tinsley, 2008;Harrison and Carslaw, 2003). The latter agrees with observations that generally negative ions possess a higher mean mobility than positive ions (Israël, 1970;Dhanorkar and Kamra, 1992;Hõrrak, 2001), i.e. on average, negative ions are of smaller sizes than positive ions.'

4. Cosmic rays are acknowledged as a significant contributor to atmospheric ionisation and hence cluster ion formation, and briefly mentioned here but are not explicitly ac- counted for in this work - it is implied that some influence is folded into the ' gamma ionising capacity' data (P9 L1) but this may only account for the proportion of activity corresponding to lower-energy photons in cosmic ray showers detectable by the spectrometer used (100-3000 keV) and would miss a substantial ' cosmic ray ionisation capacity' . If data on cosmic ray flux at the measurement site is available, this would be a very worthwhile inclusion to the paper, even if it is only discussed in a qualitative manner. For example, were there cosmic ray events or solar energetic events during the measurement period which might have influenced ionisation in a manner not ac- counted for in the Rn and gamma ionisation capacity data? What proportion of overall ionisation might be caused by sources other than the Rn and gamma ionisation cap- tured in the data presented, and is this expected to be constant throughout the study or variable? This discussion might be particularly relevant given the high latitude of the measurement site.

Thank you for the comment! Unfortunately, to our knowledge, there is no cosmic flux data available at our site.

The recorded total count rates by our gamma spectrometer were converted into dose rates in the air by a calibration factor obtained from an instrumental comparison to a pressurised ionisation chamber. The ionisation chamber basically measured the ionisation contributed by all ionizing radiation that can penetrate through the chamber wall and interact with the filled gas medium. Therefore, the gamma spectrometer could capture also muons. But for simplicity, we termed the ionising capacity derived from the dose rates recorded by the gamma spectrometer as the gamma ionising capacity. This is clarified in the revised manuscript.

However, we are aware of the possible missing of high-energy cosmic ray muons from our counting system. Either because they do not interact enough with the detector material or their light production and subsequent electrical pulses exceeding the dynamic range of the instrumentation. This point is made addressed in the revised manuscript.

Concerning solar activities, although cosmic ray intensity is anti-correlated with them, the influence contributes only a few percentages to variations in cosmic ray intensity for 2003-2006 (Moraal and Stoker, 2010;Hensen and Hage, 1994). However, as can be seen from our results, the terrestrial activity and gamma from radon decays accounted for the major contribution in the gamma ionising capacity. Therefore, even if there are remarkable solar events during these periods, they could hardly be isolated based on our dose rate data.

Below are additions and revisions made to the manuscript

In introduction

   'In the case of cosmic radiation near the ground, most of the ionisation of air is due to muons, with minor contributions from neutrons, photons and electrons (Goldhagen, 2000).'

In section 2.1.1

   'The gamma spectrometer is a scintillation-type detector using a 76 mm × 76 mm NaI(Tl) as the detection medium (Laakso et al., 2004;Hirsikko et al., 2007), which is kept at a height of 1.5 m above the ground. The device employs a ratemeter to register every single pulse with a height exceeding the background level and generates the total count rate with a time resolution of 10 min. In addition, pulse height spectra over the energy range of 100-3000 keV are recorded with a separate multichannel analyser. The total gain of the detecting system is kept constant via digital spectrum stabilization using the potassium-40 gamma peak (1460 keV) as the reference. For the determination of the ionising capacity in this work, the total count rate data was used instead of the spectral information. The recorded total count rates were converted into dose rates in the air (µSv/h) by a calibration factor obtained from an instrumental comparison to a pressurised ionisation chamber. Thus, the obtained dose rates take into account ionisation by both

gamma radiation and cosmic ray muons. However, a portion of high-energy cosmic ray muons may not be well detected by our counting system, possibly due to their weak interaction with the detector material or their light production leading to electrical pulses exceeding the dynamic range of the instrumentation.'

In section 2.2

'Therefore, the data obtained by the gamma spectrometer could be considered to represent the total gamma radiation, including the terrestrial fraction, the cosmic fraction and the fraction from radon decay. In addition, the gamma spectrometer also accounts for ionising energy from muons.'

'For conciseness and clarity, ... the ionising capacity from total dose rates recorded by the gamma spectrometer as the gamma ionising capacity ($Q\gamma$).'

5. Also, it may well be that there is not sufficient ionisation by thoron decay to justify including this as a factor in the study, and indeed this has been suggested in previous work reported from Hyytiälä (Laakso et al. (2004) ACP 4, 1933) but the possibility should at least be mentioned here, even if only to clarify it is insignificant.

**Thank you for the comment! We added description on thoron and stated that its contribution was not considered in our study and cited Laakso et al. (2004).**

**'The contributions from actinon (radon-219) and thoron (radon-220) were excluded from our study, because they are present in trace amounts naturally and have remarkably shorter half-lives (3.96 s for actinon and 55.6 s for thoron) than radon-222, which hardly permit them enough time to migrate out of the ground, especially in the case of actinon. Frozen ground and snow cover could substantially cease the transportation of thoron to the atmosphere during cold months. Even the vegetation reduces the flux of thoron from the ground to the surface air (Mattsson et al., 1996). Besides, Laakso et al. (2004) found only little contribution by thoron in the SMEAR II station to the overall radon activity.'**

6. I don't think Figure 1 in its current form is necessary. The concepts are adequately described in the text and the figure does not add much value to the description. If an 'overview' figure is deemed helpful by the authors, perhaps a schematic introducing the processes involved in

cluster ion formation and growth due to ionisation, vapour condensation, recombination etc. would be more instructive.

We appreciate the comment and suggestion of the reviewer. We added processes involved in cluster ion formation to Fig. 1. The original information on the demonstration of what are primary ions, molecular ions and cluster ions contained in Fig. 1 is retained. We intended to present clearly the relationship between primary ion, molecular ions and clusters ions and we think that a graphical illustration can better convey the message. We think that the suggestion of the reviewer would be a good addition to the figure and these processes are added now to describe better the underlying processes.

[Figure]

Figure 1. A schematic demonstration on the relationship between primary ions, molecular ions and cluster ions, as well as processes governing their formation and loss.

7. Section 3.1 and Figures 2-4 show seasonal analysis, splitting the annual cycle into 4 3-month periods. However, looking at Figure 2, the ' Spring' period shows a significant change in, in particular, the gamma ionising capacity. So, for example while the average ionising capacity (gamma + Rn) is similar to Winter when taken across this period March-May as a whole (Figure 3), the ionising capacity increases significantly during this period and also the relative importance of gamma and Rn changes. It may be worth repeating some of the analyses only for, say, mid-February to end-March, to examine this specific period where Rn contributes the largest fraction to overall ionising capacity across the whole year, as I wonder whether features are being diluted and missed by taking the average over a period in which large changes in these parameters occur.

**Thank you for the comment! The feature was blurred slightly by the 3-month average window, as can be seen from the figures below. There was a weak diurnal cycle in the min-Feb to end-March period and the ratio of medians in this period to medians in spring and in winter are 1.07 and 1.09, respectively. The contribution by radon in this mid-Feb to end-March period fell in between those in spring and winter. These hidden features are not very remarkable and, therefore, the choice of the four 3-month periods can be still considered representative.**

[Figure]

[Figure]

8. P4 L25 Please clarify whether '  during 2003-2006'  means whole-year data collection for these years, or if not, in which months data collection/use began/ended.

**Thank you for the comment! The data were from our long-term routine measurements. There were small gaps every now and then due to maintenance activities. The ion spectrometer was introduced in 2003 and therefore there were about 2-month data missing in the beginning of 2003. But otherwise, we had almost full coverage of data over these four years. For the ionising radiation measurements, we had almost full data coverage in 2005 and 2006 for radon data and in 2003 and 2006 for total gamma. We mainly had gaps in ionising radiation measurements in the latter half of year 2004. However, gathering up the four-year data, we have a very large dataset and have data for every single day-of-year. Since we focused mostly on the general trends shown by the medians of the data, some gaps in the data could hardly bias our interpretation.**

**Information about data coverages is added in the revised manuscript**

**In section 2.1**

  **'For the study period of 2003-2006, the data availability for air ions, particles, radon and gamma was 91%, 99%, 71% and 76%, respectively, allowing the coverage of every single day-of-year by each dataset.'**

9. P6 I note also that APi-TOF may not provide direct information on the actual electrical mobility of atmospheric cluster ions, as encountered in environmental measurements using air ion spectrometers, because of the evaporation of clustering water molecules during the APi-TOF

measurement.

**The low pressure used in mass spectrometers could indeed cause loss of water and other loosely attached molecules on the sampled clusters. Such breaking-up of clusters is more likely to happen in mass spectrometers than in air ion spectrometers. APi-ToF is a mass spectrometer for study the mass distribution of atmospheric clusters, but it provides no direct information on mobility. Although some information about the mobility of the sample clusters can be obtained via a comparison with an ion spectrometer, like the work down by Ehn et al. (2011), the mass-mobility relationship has some dependency on the choice of model, as shown by Ehn et al.**

10. P7 Please state the voltage scan time for the BSMA in this work (it looks like it appears in the legend to Figure 8 but should be specified in the main text as well).

    **We added this information in the revised manuscript as follows**

    **'A full measurement cycle scanning through the mobility range for both polarities takes 10 min.'**

11. P8 L11 Please clarify your justification for using only 'O horizon' soil data.

    **We add the reasoning for using O horizon soil data in the revised manuscript as follows**

    **'Measurements on soil temperature and soil volumetric water content were described by Pumpanen et al. (2003) and Ilvesniemi et al. (2010). Only the organic horizon data (5 cm depth, above the mineral layer (Pumpanen et al., 2003)) were used in this work. The organic horizon is in direct contact with the atmosphere, the condition of which exerts the primary influence on radon exhalation.'**

12. P4 L25 "Monitoring devices for" not "The monitoring devices of the"

    **Thank you for the correction! Changes are made accordingly**

13. P5 L15 "described" not "descripted"

**Thank you for spotting out our typo! It is corrected in the revised manuscript.**

14. P8 L8 "Time-Domain Reflectometer" or "…Reflectometry (TDR) device."

**Thank you for pointing this out! However, since the instrumentation for soil measurement has been described carefully by Pumpanen et al. (2003) and Ilvesniemi et al. (2010), we decided to remove the details from our paper, as suggested by the other reviewer. We add citations to those published works for readers who would like to know more about the measurement.**

15. P12 L10 "examine" not "exam"

**Thank you for the correction! It is changed accordingly.**

16. P12 L22 what is meant by "shading the variability"?

**We intended to say that we investigated the connection between the radon ionising capacity and RH after eliminating the influence brought by MLH, temperature and wind on the radon ionising capacity. And it is rephrased in the revised manuscript as follows**

**'··· no clear correlation between the radon ionising capacity and RH was found based on our dataset after ruling out the variability in the radon ionising capacity brought by MLH, temperature and wind.'**

17. P13 L9-11 move text "to the measured snow depth data" to L9: "exponential fitting … could represent" or otherwise reword, to clarify what is relationship is being probed.

**Thank you for pointing out this! We reformulated the sentence as follows**

**'···the constant term in the exponential fitting to the measured snow depth data, being about 3 $cm^{-3} s^{-1}$, could represent an approximation of the contribution by cosmic radiation to the ionising capacity.'**

18. P14 L9 "blooming" not "booming"? P20 L9 "localised" not "focalised"? P20 L26 "level out" not "level up" P22 L2 "support" not "supports"

**Thank you for these suggestions! Followings are the changes made in the revised manuscript: 'booming season' to 'growth season' , 'focalised' to 'localised' , 'level up' to 'level out' and 'support' to 'supports' .**

19. P22 L19-22 the full citation now appears to be available online, please amend.

**Thank you for pointing this out! The full citation is added in the revised manuscript.**

20. P33 L4 "0-4 cm" not "-4-0 cm". P34 Fig 2, both y-axes "ionising" not "ioning"

**Thank you for spotting these typos out! They are corrected accordingly.**

21. P43 L3 "T was below" not "T below"

**Thank you! It is corrected accordingly.**

References

Aplin, K. L., and Harrison, R. G.: A computer-controlled Gerdien atmospheric ion counter, Review of Scientific Instruments, 71, 3037, 10.1063/1.1305511, 2000.
Canton, J.: Electrical Experiments, with an Attempt to Account for Their Several Phaenomena; Together with Some Observations on Thunder-Clouds, Phil. Trans., 48, 350-358, 1753.
Cumeras, R., Figueras, E., Davis, C. E., Baumbach, J. I., and Gracia, I.: Review on ion mobility spectrometry. Part 1: current instrumentation, Analyst, 140, 1376-1390, 10.1039/c4an01100g, 2015.
Dhanorkar, S., and Kamra, A. K.: Relation between electrical conductivity and small ions in the presence of intermediate and large ions in the lower atmosphere, J. Geophys. Res., 97, 20345-20360, 1992.
Ehn, M., Heikki, J., Siegfried, S., E., M. H., Alessandro, F., Mikko, S., Tuukka, P., Veli-Matti, K., Hannes, T., Aadu, M., Sander, M., Urmas, H., Markku, K., and Douglasx, W.: An instrumental comparison of mobility and mass measurements of atmospheric small ions, Aerosol Science and Technology, 45, 522-532, 10.1080/02786826.2010.547890, 2011.
Gerdien, H.: Die absolute messung der specifischen leitfähigkeit und der dichte des verticalen

leitungsstromes in der atmosphäre., Terrestrial Magnetism and Atmospheric Electricity, X, 65-74, 1905.

Goldhagen, P.: Overview of aircraft radiation exposure and recent er-2 measurements, Health Physics, 79, 526-544, 2000.

Harrison, R. G., and Carslaw, K. S.: Ion-aerosol-cloud processes in the lower atmosphere, Rev. Geophys., 41, 10.1029/2002rg000114, 2003.

Hensen, A., and Hage, J. C. H. v. d.: Parameterisation of cosmic radiation at sea level, J. Geophys. Res., 99, 10693-10695, 1994.

Hirsikko, A., Paatero, J., Hatakka, J., and Kulmala, M.: The $^{222}$Rn activity concentration, external radiation dose and air ion production rates in a boreal forest in Finalnd between March 2000 and June 2006, Boreal Env. Res., 12, 265 - 278, 2007.

Hirsikko, A., Nieminen, T., Gagné, S., Lehtipalo, K., Manninen, H. E., Ehn, M., Hõrrak, U., Kerminen, V. M., Laakso, L., McMurry, P. H., Mirme, A., Mirme, S., Petäjä, T., Tammet, H., Vakkari, V., Vana, M., and Kulmala, M.: Atmospheric ions and nucleation: a review of observations, Atmospheric Chemistry and Physics, 11, 767-798, 10.5194/acp-11-767-2011, 2011.

Hõrrak, U.: Air Ion Mobility Spectrum at a Rural Area, PhD, Institute of Environmental Physics, University of Tartu, Tartu, Estonia, 81 pp., 2001.

Ilvesniemi, H., Pumpanen, J., Duursma, R., Hari, P., Keronen, P., Kolari, P., Kulmala, M., Mammarella, I., Nikinmaa, E., Rannik, U. I., Pohja, T., Siivola, E., and Vesala, T.: Water balance of a boreal scots pine forest, Boreal Env. Res., 15, 375-396, 2010.

Israël, H.: Atmospheric electricity, Vol. I, Israel Program for Scientific Translations, Jerusalem, 1970.

Junninen, H., Ehn, M., Petäjä, T., Luosujärvi, L., Kotiaho, T., Kostiainen, R., Rohner, U., Gonin, M., Fuhrer, K., Kulmala, M., and Worsnop, D. R.: A high-resolution mass spectrometer to measure atmospheric ion composition, Atmos. Meas. Tech., 3, 1039-1053, 10.5194/amt-3-1039-2010, 2010.

Kruis, F. E., Fissan, H., and Peled, A.: Synthesis of nanoparticles in the gas phase for electronic, optical and magnetic applications—a review, J. Aerosol Sci., 29, 511-535, 1998.

Ku, B. K., and de la Mora, J. F.: Relation between Electrical Mobility, Mass, and Size for Nanodrops 1–6.5 nm in Diameter in Air, Aerosol Science and Technology, 43, 241-249, 10.1080/02786820802590510, 2009.

Kulmala, M., Hõrrak, U., Manninen, H. E., Mirme, S., Noppel, M., Lehtipalo, K., Junninen, H., Vehkamäki, H., Kerminen, V.-M., Noe, S. M., and Tammet, H.: The legacy of Finnish–Estonian air ion and aerosol workshops, Boreal Env. Res., 21, 181-206, 2016.

Laakso, L., Petäjä, T., Lehtinen, K. E. J., Kulmala, M., Paatero, J., Hõrrak, U., Tammet, H., and Joutsensaari, J.: Ion production rate in a boreal forest based on ion, particle and radiation measurements, Atmos. Chem. Phys., 4, 1933–1943, 2004.

Laskin, A., Laskin, J., and Nizkorodov, S. A.: Mass spectrometric approaches for chemical characterisation of atmospheric aerosols: critical review of the most recent advances, Environmental Chemistry, 9, 163, 10.1071/en12052, 2012.

Manninen, H. E., Petäjä, T., Asmi, E., Riipinen, I., Niemnen, T., Mikkilä, J., Hõrrak, U., Mirme, A., Mirme, S., Laakso, L., Kerminen, V.-M., and Kulmala, M.: Long-time field measurements of charged and neutral clusters using Neutral cluster and Air Ion Spectrometer (NAIS) Boreal Env. Res., 14, 591-605, 2009.

Mattsson, R., Paatero, J., and Hatakka, J.: Automatic alpha/beta analyser for air filter samples - absolute determination of radon progeny by pseudo-coincidence techniques, Radiation protection dosimetry, 63, 133-139, 1996.

Mirme, A., Tamm, E., Mordas, G., Vana, M., Uin, H., Mirme, S., Bernotas, T., Laakso, L., Hirsikko, A., and Kulmala, M.: A wide-range multi-channel Air Ion Spectrometer, Boreal Environment Reseach, 12, 247-264, 2007.

Moraal, H., and Stoker, P. H.: Long-term neutron monitor observations and the 2009 cosmic ray maximum, Journal of Geophysical Research: Space Physics, 115, n/a-n/a, 10.1029/2010ja015413, 2010.

Pumpanen, J., Ilvesniemi, H., Perämäki, M., and Hari, P.: Seasonal patterns of soil $CO_2$ efflux and soil air $CO_2$ concentration in a Scots pine forest: comparison of two chamber techniques, Global Change Biol., 9, 371-382, 2003.

Tammet, H.: The aspiration method for the determination of atmospheric-ion spectra, Israel Program for Scientific Translations, Jerusalem, 1970.

Tammet, H.: Continuous scanning of the mobility and size distribution of charged clusters and nanometer particles in atmospheric air and the Balanced Scanning Mobility Analyzer BSMA, Atmos. Res., 82, 523-535, 10.1016/j.atmosres.2006.02.009, 2006.

Tammet, H.: Symmetric Inclined Grid Mobility Analyzer for the measurement of charged clusters and fine nanoparticles in atmospheric air, Aerosol Sci. Tech., 45, 468-479, 10.1080/02786826.2010.546818, 2011.

Tinsley, B. A.: The global atmospheric electric circuit and its effects on cloud microphysics, Rep. Prog. Phys., 71, 066801, 10.1088/0034-4885/71/6/066801, 2008.

Vojtek, T., Skoupil, T., Fiala, P., and Bartu--ek, K.: Accuracy of Air Ion Field Measurement, Electromagnetics Research Symposium, Cambridge, USA, 2006.

Wilson, C. T. R.: Investigations on Lightning Discharges and on the Electric Field of Thunderstorms, Philos. Trans. R. Soc. London A., 221, 73-115, 1921.

22.

---

## Author Response (AR1)

Helsinki, Sept. 30, 2016

Dear Editor,

Our manuscript has been carefully revised according to comments received from the two reviewers.

As suggested by referee #1, we changed the title of the manuscript to 'How do air ions reflect variations in ionising radiation in the lower atmosphere in a boreal forest?'. The detailed reply to each comment presented by the reviewers can be found in the enclosed responses. This file also contains a track-change version of the revised manuscript at the end, where changes are marked in red. In addition to revisions based on reviewers' suggestions, we also made a few technical modifications and corrections and these are also outlined in red colour.

We look forward to your reply!

Yours sincerely,

Xuemeng Chen
(xuemeng.chen@helsinki.fi)

Response to Referee 1

We would like to deliver our gratitude to referee #1 for the detailed and thorough review of our work. Below are our responses to the critical points presented by the referee. For each comment of the referee, our reply consists of a general response and an indication of changes in the manuscript following the guidelines.

1. The introductory material is narrow, and implies that (a) this is a topic which has hardly been worked on and (b) there is limited understanding of how radioactivity makes ions. Since radioactivity and cosmic rays were discovered through their ionisation of the atmosphere at least a hundred years ago, these implications are incorrect. The authors should be referring to some historical material (e.g. the Irish work of Nolan and collaborators) or, at a minimum, a historical review, and the work on ion measurements from other groups in the late twentieth century, to explain how their study progresses the research area.

**Thank you for the comment! We broadened our introduction concerning the historical works on air ion study and instrumentations used for characterising air ions. This part of the manuscript is elaborated as follows**

**'Air ions were historically concerned in the discipline of atmospheric electricity (Israël, 1970), because their flow in the electric field of the earth-atmosphere system serves as the measureable conduction current in the atmosphere (Wilson, 1921;Tinsley, 2008;Harrison and Carslaw, 2003). The interest in atmospheric electricity could be traced back to the early 18[th] century when thunderstorms were suggested to be electrical phenomena (Herbert, 1997). However, only when Benjamin Franklin proposed the idea to draw electricity down from lightning in 1752, this theory was confirmed and the study of atmospheric electricity became popular (Herbert, 1997;Tinsley, 2008). Early efforts in this field were substantially invested into understanding lightning and electrification of clouds (e.g. (Canton, 1753a, b;Franklin, 1751)), even though there were reports on observations of atmospheric electricity under fair weather conditions (Canton, 1753a;Read, 1792;Bennett and Harrison, 2007). But why the air is conductive could not be explained. Meanwhile, Charles-Augustin de Coulomb observed gradual discharge of a well-insulated electroscope around 1785 and he attributed his observation to the contact of suspending particles present in the air (Walter, 2012;Angelis, 2014). This phenomenon was reproduced by Michael Faraday half a century later in 1835 (Angelis, 2014). Thanks to the further improvement of the electroscope by William Thomson and Lord Kelvin (Angelis, 2014;Flagan, 1998), Crookes (1878) could find that the discharge**

rate of an electroscope decreases with a decreasing air pressure, suggesting that it is the air inside the instrument that manipulates the discharge. Yet, the reasoning remained undisclosed, until the discovery of radioactivity by Wilhelm Röntgen, Henri Becquerel and Marie and Peirre Curie enabled Julius Elster and Hans Geitel from Germany and Charles Thomson Rees Wilson from Scotland to relate the spontaneous discharge of the electroscope to ionisation of the air by radioactive sources (Angelis, 2014;Carlson and De Angelis, 2011). Therefrom, the importance of air ions in the atmosphere emerged. Contemporaneously, the interest of Joseph John Thomson, director of the Cavendish Laboratory, in the charge carriers produced by ionising radiation motivated the development of instrumentations for measuring electrical charges in gases, leading to various valuable outcomes, e.g. the cloud chamber designed by C. T. R. Wilson, as well as techniques for measuring ion mobility by Ernest Rutherford and John Zeleny and for studying gaseous ion diffusion by John Sealy Townsend (Flagan, 1998;Robotti, 2006). These works laid the theoretical and instrumental foundation for later aerosol studies. The experimental results from C. T. R. Wilson's cloud chamber measurements in 1895 and 1899 on the influence of ionising radiation on the formation of cloud droplets brought interests of air ions into the atmospheric aerosol community. Inspired by these early works, advancements in atmospheric aerosol studies progressed both instrumentally and theoretically over the century (e.g. Tammet 1970; Nolan, 1924; Hogg, 1939; Hewitt, 1957; Tammet, 2006; Reischl, 1991; Rosell-Llompart and Fernández de la Mora, 1993; Aplin and Harrison, 2000; Mason and McDaniel, 1988; Millikan, 1923; Hinds, 1999; Tammet, 1995).

Devices employed in air ion studies comprise different types of aspiration condensers, ion mobility spectrometers (IMS) and mass spectrometers (Hirsikko et al., 2011;Tammet, 1970;Laskin et al., 2012;Cumeras et al., 2015). Notably, modern key instruments for field observations of air ions are mainly aspiration contender-based devices and mass spectrometers, such as the Gerdien counter - an integral aspiration condenser (Gerdien, 1905;Aplin and Harrison, 2000;Vojtek et al., 2006), ion spectrometers designed by Airel Ltd – single or multiple channel aspiration condensers (Tammet, 2011, 2006;Mirme et al., 2007;Manninen et al., 2009;Kulmala et al., 2016) and the atmospheric pressure interface time-of-flight mass spectrometer (APi-ToF) (Junninen et al., 2010). While aspiration condensers provide information on the concentration and mobility of charge carriers, mass spectrometers reveal mainly the chemical properties of them. The IMS, however, has a limited

application in studying ambient ions, due to difficulties in spectrum interpretation (Hirsikko et al., 2011). The purpose of these instrumentations is not only limited to air ion or air conductivity observations, but also for nano-material synthesis (Kruis et al., 1998) and for the improvement of the fundamental understanding on the relationship between mobility, mass and size (Ku and Fernández de la Mora, 2009).'

2. The motivation of the paper needs to be refocused towards the work that is described, which is a far more sophisticated study of the interaction between radioactivity and the properties of atmospheric ions than the introduction suggests.

   **We appreciate the reviewer's comment. We refocused the motivation of our paper to better match the work we presented. It is as follows**

   **'… Such deficiencies prompt the motivation of this work to examine how variations in ionising radiation are reflected in observed air ions based on ambient measurement. The aims of this study are 1) to identify the key factors responsible for the variability in ionising radiation and in observed air ion concentrations, 2) to reveal the linkage of observed air ions to variations in ionising radiation and 3) to provide an in-depth analysis on the effects of ionising radiation on air ion formation. We will first introduce factors that cause the seasonal and diurnal variability in ionising radiation and air ions, and then exposit the connection of observed air ions to variations in ionising radiation and the influence of different atmospheric conditions on this relationship. '**

3. The title of the paper also implies a much more basic study than is actually carried out and might be better changed to something like, "Effects of ionising radiation on ion mobility in a range of atmospheric conditions" (this is just a suggestion and should not be taken literally, but I hope you understand my point).

   **Thank you for the comments! We decided to change our title to 'How do air ions reflect variations in ionising radiation in the lower atmosphere in a boreal forest?' to better stand for the content of our paper.**

4. A second aspect of the paper that indicates lack of awareness of other studies is that the authors present their own definitions for terms that are already precisely defined. For example, they introduce a term called "ionising capacity" which

to all intents and purposes appears to be identical to what the rest of the community already calls "ionisation rate", since it has the same units and even the same symbol. The only justification I can think of for bringing in this new terminology would be if the ionising capacity were a theoretical maximum amount of ionisation, given the energies and activities of the particles involved, which may not be the same as the actual ionisation rate. How- ever, since the authors make no attempt to justify their new definition, and also, the only way to measure the "true" ionisation rate would be through detailed ion measurements of the sort made by the authors, making the distinction between "capacity" and "rate" doesn't seem particularly helpful. Some of the energy of the radioactive particles will be used for excitation and not ionisation, but I would imagine that this is a relatively small fraction, and it is not unreasonable to assume that all the energy lost by the radioactive particles goes into ionisation. I recommend that the authors remove the references to and definition of "capacity" and simply talk about ionisation rate for consistency with other work. They could add a caveat mentioning energy loss to excitation if this concerns them.

**Thank you for the comments! Indeed, we introduced a new term, ionising capacity. We intended to distinguish between the actual ionisation rate and the theoretical maximum ionisation rate based on calculation.**

**The ionisation rate is a measure of the rate of successful production of ion pairs by ionisation, which is self-evident. However, the usage of this term has been extended beyond its scope vaguely sometimes. There could be energy loss due to excitation. It seems that the reviewer has gotten this point. However, whether this energy expenditure occupies a minor share or not is something that remains unknown and needs further quantification.**

**Another thing that needs to be aware of is that the ionisation rate should be a universal quantity determined by natural processes, which however, cannot be measured directly yet. Our determination of the rate is based on ionising radiation measurements and the assumption that 34 eV is needed for producing one ion pair. There are uncertainties associated with measurements. Besides, how well can the 34 eV characterise the natural ionisation process at different heights of the atmosphere and at different spatial locations requires further investigation.**

**For the reasons stated above, we would like to keep the term, ionising**

capacity, and to encourage the research community to be cautious with the term usage in describing 'the rate of producing air ions'. We added a better description of the ionising capacity to justify its connection to and difference from the ionisation rate in the revised manuscript.

'The ionising capacity can be viewed as a measure of the theoretical maximum ionisation rate, which however, may not well capture the true ionisation rate due to uncertainties in ionising radiation measurements, possible energy dissipation of ionising radiation in excitation and the invalidity concern associated with the use of 34 eV per production of an ion pair at near ground level in our calculation'

5. The wheel is reinvented again in figure 1. Many years of work in the early twentieth century went into defining cluster ions, small ions, large ions, etc. If the authors are going to come up with their own definitions they must explain why they are needed. In particular, the distinction between primary and molecular ions seems arbitrary, since the primary ions N2+ and O2+ are technically molecular ions anyway, and the ions you call molecular ions I thought were unstable in the atmosphere and cluster immediately.

Thank you for the comment! It is true that there exist definitions on clusters ions, small ions and large ions from historical works. For example, H. Israël (1970) wrote in the famous 'Atmospheric electricity' book that 'An electron, detached from a molecular or atomic bond, cannot exist freely in air at normal temperature and pressure, but readily attaches itself to a neutral atom or molecule. However, even these molecular or atomic ions cannot remain stable in atmospheric air under normal conditions, and consequently surround themselves with a number of neutral molecules and form clusters of approximately 10-30 molecules. Only then do they reach a certain stability in the form of so-called "small ions."'

The information in our Fig. 1 does not contradict with what H. Israël explained, rather we followed his description. H. Israël considered that a 'small ion' is a charged cluster that has reached certain stability threshold and he was aware of the existence of atomic and molecular ions, though they are not stable in the atmosphere to be readily clustered.

Theoretical studies have shown that only after reaching the critical size, clusters are stable enough thermodynamically for growing bigger, which is

likely the 'Israël threshold' for charged clusters to be called small ions. The critical size was found to be around 1.5±0.3 nm for atmospheric nucleation (Kulmala et al., 2013). In our study, we concentrated on ions in the size range between 0.8 and 1.7 nm, which we called the cluster size range. We discussed in the paper how ions in this size range are possibly formed from smaller ions (primary ions and molecular ions) and the possible underlying dynamic processes. For this purpose, it is necessary to introduce what are primary ions, molecular ions and cluster ions, and to make a distinction between them.

The boundary between primary ions and molecular ions does not exist explicitly. It is the origin of the ions that tells what ions they are. If a charged single molecule is formed from ionisation directly, it is also a primary ion. As the reviewer wrote, primary ions N2+ and O2+ can also be classified as molecular ions intrinsically. However, if the molecule gets its charge from other charged species in the atmosphere, it is only a molecular ion, but not a primary ion. We would like to show Fig. 1 instead of only explaining in plain text to convey the message clearly to our readers that the scope of primary ions and that of molecular ions do overlap in certain aspects considering the origin of the charge. As for the stability and lifetime of molecular ions, certain big molecular ions may exist in the atmosphere, which have been demonstrated by the recent application of APi-ToF MS in field measurement (Ehn et al., 2010).

We are also aware of the possible presence of molecular ions in our cluster size range as stated in the manuscript. There is no such a clear size boundary to distinguish between molecular ions and cluster ions. To avoid misleading of readers by what we called 'cluster ions' in our analysis, we considered it important to demonstrate what truly distinguishes a molecular ion from a cluster ion. And this purpose is served by Fig. 1.

By taking the comment from the other reviewer into consideration, however, we added underlying processes in cluster ion formation into Fig. 1. Below is the revised figure.

[Figure]

**Figure 1. A schematic demonstration of the relationship between primary ions, molecular ions and cluster ions, as well as processes governing their formation and loss.**

6. Fourthly, the authors do not seem aware of the known theoretical relationships between ionisation rate and ion concentration. The paper they cite by Harrison and Carslaw (2003) contains a good introduction to this theory. The lack of theoretical awareness is particularly apparent when looking at figure 13, which seems to be a very nice demonstration of the ion balance equation in the recombination limit. I've seen similar plots before (e.g. in Aplin and Harrison, Rev Sci Instrum 2000), but this is by far the best data, however the authors do not place this work in context. They need to talk about the ion balance equation and compare their work to those of other groups; this will demonstrate the strength of their data.

**We appreciate the reviewer's comment! The flattening out of cluster ion concentrations at higher ionising capacities in Fig. 13 is related to the fact that recombination becomes the determining process. This was said in the manuscript. However, this feature cannot be explained by the ion balance equation at the recombination limit alone. As given by Harrison and Carslaw (2003), the steady state ion concentration at the recombination limit is $n = $ sqrt($q/\alpha$), where $q$ is the ionising capacity in our case and $\alpha$ is the**

recombination coefficient being about 1.6e-6 cm3s-1. However, as can be seen from $n = \mathrm{sqrt}(q/\alpha)$, $n$ will increase monotonically with an increasing $q$, but does not level out. Even with the addition of the ion-aerosol interaction term $\beta N n$, which can also be written as the product of the condensational sink and ion concentration – $CSn$, then $n = \mathrm{sqrt}(q/\alpha+(CS/2/\alpha)^2)-CS/2/\alpha$, the ion concentration still does not flatten out. This flattening-out could result from the synergic effect of different complicated processes, especially because we are dealing with ambient data. However, one of the reasons may be related to the availability of vapour sources or neutral clusters. But this speculation needs to be verified with further knowledge on the roles of different key chemical species participating in cluster ion and neutral cluster formation and diurnal variations in their atmospheric concentrations, which however are beyond the scope of this work.

We would like to thank the reviewer for suggesting the interesting work of Aplin and Harrison (2000). We suppose that the reviewer meant for Fig 4 in Aplin and Harrison (2000), where a relationship can be seen between the ionising radiation and air ion concentration. However, the information conveyed by this figure may not be comparable with our observation. In their work, they observed a continuous increase in the ion concentration, even if the ionising radiation showed a levelling-out. But in our case, it is the opposite that we saw suppressed cluster ion production along with a continuous increase ionising capacity.

7. The paper presents a huge amount of data is presented, in a long main text, but the conclusions are brief bordering on the obvious. The conclusions need to present a synthesis of the data and put it into context in the conclusions. It is also not clear why the results are important – for example the last sentence of the entire paper recommends that instruments that can measure sub-0.8 nm ions are needed – why?

We improved the conclusions according to the reviewer's comments. We addressed the importance of our results and the reason why instruments for sub-0.8 nm ion measurement are needed. The revised conclusions are as follows

'In this work, diurnal and seasonal cycles in ionising radiation were presented and key influencing factors responsible for these features were overviewed in order to investigate how observed air ions respond to these

variations and to improve our understanding on air ion formation. To assist the analysis, a term, ionising capacity, was introduced to capture patterns in ionising radiation. The ionising capacity was determined theoretically as the potential maximum production rate of ion pairs in the atmosphere by ionising radiation, based on the assumption that an ion pair is produced upon every 34 eV energy dissipation of ionising radiation. The data used in this study were collected from ambient measurements during 2003-2006 from a boreal forest site in southern Finland. In our analysis, the accounted ionising radiation is composed of energy from alpha and beta decays of Rn-222 and accompanying gamma radiation, energy contained in the gamma radiation from terrestrial origins and gamma radiation released from the interactions between cosmic rays and air molecules. Variations in the ionising capacity were primarily related to boundary layer development, soil conditions, snow accumulation and the origin of air masses.

Although ionising radiation is known to be responsible for air ion production, patterns in the measured air ion concentration in the cluster size range (0.8-1.7 nm) did not exhibit a highly comparability to those in the ionising capacity, due to modifications of air ion properties exerted by different dynamical processes and chemical reactions during the evolution of charges in the atmosphere. Nevertheless, the connection of air ions to ionising radiation was seen for air ions in the lowest detected size band (0.8-1 nm) of the cluster size range (0.8-1.7 nm). The evolvement of these 0.8-1 nm ions with time to larger sizes in the cluster size band was also identified, affirming the primary role of ionising radiation in the production of air ions in the lower atmosphere. Yet, atmospheric conditions, such as temperature, humidity and pre-existing aerosol particles, brought complications into this relationship. By carefully constraining data to conditions of a similar meteorology, seasonality, diurnality and amount of background aerosol particles, a strong dependency of total cluster ion concentrations on the ionising capacity was identified on new particle formation (NPF) days. However, the linkage was not visible on non-event days. These observations may suggest that charges, after being born, underwent different processes on NPF days and non-event days and possibly indicate also that the transformation of newly formed charges to cluster ions occurred faster on NPF days than on non-event days. These results could help to advance our understanding on the role of ions in atmospheric new particle formation.

However, to obtain further insights into the fate of charges created by

ionising radiation in the atmosphere, i.e. ion balance, and into the role of air ions in the atmospheric new particle formation process, it is crucial to understand the transformation process of electric charges into detectable air ions. For this purpose, knowledge on the number size distribution of air ions smaller than 0.8 nm is of necessity. Additionally, theoretical understanding on the formation mechanisms of cluster ions from molecular ions needs to be deepened. Conjointly, also advancing instrumental development for the detection of sub-0.8 nm ions could be worthy of being brought onto the agenda.'

8. The results on new particle formation are novel and very interesting, but can you discuss their implications?

Thank you for the comment! The better dependency of cluster ion concentrations on the ionising capacity observed on NPF event days than that on non-event days may imply that charges on NPF event days did not undergo significant transformations after being produced via ionisation by ionising radiation to become cluster ions. Therefore, it is possible to trace the linkage between the cluster ion concentration and ionising capacity statistically. However, this connection was very poorly preserved on non-event days, possibly because after being created, charges went through a too complicated series of modifications before reaching the cluster sizes.

In other words, these results could provide us some hints on what charges have experienced in the atmosphere after formation and possibly also on the relative duration of the processes faced by newly formed charges before they become cluster ions. It could be that charges became cluster ions in a shorter time scale on NPF event days than on non-event days, which allows those cluster ions on NPF event days to retain some features in connection to ionising radiation.

These results could help advance our understanding on the role of ions in atmospheric new particle formation. Apart from this, these results could also be beneficial in model development aiming for insights into the new particle formation process.

These implications are added in the revised manuscript as follows

In the discussion of Fig. 13

'	In addition, the better dependency of the cluster ion concentration on the ionising capacity observed on NPF event days than on non-event days may also imply that charges, after being produced via ionisation by ionising radiation, on NPF event days did not undergo significant transformations to become cluster ions. Therefore, the linkage between the cluster ion concentration and ionising capacity was traceable statistically. However, this connection was very poorly preserved on non-event days, possibly because after being created, charges went through a too complicated series of modifications before reaching the cluster sizes. These observations may also provide some indirect measure of the relative duration of dynamic processes faced by newly formed charges before they become cluster ions under different atmospheric conditions: it might suggest that charges became cluster ions in a shorter time scale on NPF event days than on non-event days and as a consequence, the cluster ions on NPF event days are enabled to retain some features in connection to ionising radiation.'

In the conclusion

'These observations may suggest that charges, after being born, underwent different processes on NPF days and non-event days and possibly indicate also that the transformation of newly formed charges to cluster ions occurred faster on NPF days than on non-event days. These results could help to advance our understanding on the role of ions in atmospheric new particle formation.'

9. A final scientific point is that the authors do not adequately explain at the beginning of the paper that they are omitting to measure most of the cosmic rays and that they usually contribute about 20% of the ionisation rate. They are mentioned in the context of high-energy gamma radiation, but there are muons and electrons too which are not mentioned. Later in the text, it is implied from the residuals that the cosmic ionisation rate is 3 /cc/s, and it is argued that because the "textbook" rate is 2 /cc/s, then one third of the charge created is lost before even becoming atmospheric ions. This seems totally speculative; surely it is much more likely that the fit/measurement errors and the high latitude of the Finnish measurement site and perhaps solar activity could account for the discrepancy. It would be possible to carry out a more quantitative analysis here.

**We agree with the reviewer that 2 or 3 /cc/s and the text about**

recombination are highly speculative. It can well fall (and most probably does fall) into the uncertainty budget. The discussion is revised.

[Figure]

As shown in the figure above (Goldhagen, 2000), the photon and electron contributions from cosmic ray at Hyytiälä altitude is about 10 % of the muon contribution, the same with neutrons, i.e. it is the muons that count.

The recorded total count rates by our gamma spectrometer were converted into dose rates in the air by a calibration factor obtained from an instrumental comparison to a pressurised ionisation chamber. The ionization chamber basically measured the ionization contributed by all ionizing radiation that can penetrate through the chamber wall and interact with the filled gas medium. Therefore, the gamma spectrometer could capture muons.

But for simplicity, we termed the ionising capacity derived from the dose rates as the gamma ionising capacity. This is clarified in the revised manuscript.

However, a portion of high-energy muons may not be detected well by our counting system. This is either because they do not interact enough with the detector material, or because their light production causes subsequent electrical pulses exceeding the dynamic range of the instrumentation.

Descriptions and revisions on these aspects are as follows

In introduction

'In the case of cosmic radiation near the ground, most of the ionisation of air is due to muons, with minor contributions from neutrons, photons and electrons (Goldhagen, 2000).'

In section 2.1.1

'The gamma spectrometer is a scintillation-type detector using a 76 mm × 76 mm NaI(Tl) as the detection medium (Laakso et al., 2004;Hirsikko et al., 2007), which is kept at the height of 1.5 m above the ground. The device employs a ratemeter to register every single pulse with a height exceeding the background level and generates the total count rate with a time resolution of 10 min. In addition, pulse height spectra over the energy range of 100-3000 keV are recorded with a separate multichannel analyser. The total gain of the detecting system is kept constant via digital spectrum stabilization using the potassium-40 gamma peak (1460 keV) as the reference. For the determination of the ionising capacity in this work, the total count rate data were used instead of the spectral information. The recorded total count rates were converted into dose rates in the air (µSv/h) by a calibration factor obtained from an instrumental comparison to a pressurised ionisation chamber. Thus, the obtained dose rates take into account ionisation by both gamma radiation and cosmic ray muons. However, a portion of high-energy cosmic ray muons may not be well detected by our counting system, possibly due to their weak interaction with the detector material or their light production leading to electrical pulses exceeding the dynamic range of the instrumentation.'

In section 2.2

'Therefore, the data obtained by the gamma spectrometer could be considered to represent the total gamma radiation, including the terrestrial fraction, the cosmic fraction and the fraction from radon decay. In addition, the gamma spectrometer also accounts for ionising energy from muons.'

'For conciseness and clarity, ... the ionising capacity from total dose rates recorded by the gamma spectrometer as the gamma ionising capacity ($Q\gamma$).'

In section 3.2.2

'According to Paatero et al. (2005), the constant term in the exponential fitting to the measured snow depth data, being about 3 $cm^{-3}$ $s^{-1}$, could represent an approximation of the contribution by cosmic radiation to the ionising capacity. A rate of 2 $cm^{-3}$ $s^{-1}$ has been generally accepted as the ionisation rate of cosmic radiation for the production of small ions at sea level (Hensen and Hage, 1994). The cosmic contribution to the ionising capacity determined from our measured total dose rates with the exponential fittings was close to this value. The discrepancy comes likely from uncertainties involved in the mathematical fitting and measurement as well as possible spatial variations in the cosmic radiation ionisation rate and solar activity influence.'

10. P3 L12 "generated electric charges from . . . by the derivatives of them" does not make sense

Thank you for pointing this out! The sentence has been rephrased as

'Due to the atmospheric abundance of nitrogen (N2) and oxygen (O2), their derivatives are the initial carriers of electric charges generated from the ionisation process.'

11. P4 L7-21 Please revise this paragraph in line with the comments above

Thank you for the comment! We revised this paragraph as follows

'Although it is known that ionising radiation creates ion pairs via ionisation in the atmosphere (Flagan, 1998;Harrison and Carslaw, 2003;Israël, 1970),

except for a few attempts (Hirsikko et al., 2007;Laakso et al., 2004), minor efforts have been invested in understanding the connection between ionising radiation and observed air ions in the lower atmosphere. Moreover, there is a lack of quantification on the underlying processes. Such deficiencies prompt the motivation of this work to examine how variations in ionising radiation are reflected in observed air ions based on ambient measurement. The aims of this study are 1) to identify the key factors responsible for the variability in ionising radiation and in observed air ion concentrations, 2) to reveal the linkage of observed air ions to the variations in ionising radiation and 3) to provide an in-depth analysis on the effects of ionising radiation on air ion formation. We will first introduce factors that cause the seasonal and diurnal variability in ionising radiation and air ions, and then exposit the connection of observed air ions to variations in ionising radiation and the influence of different atmospheric conditions on this relationship. To assist our analysis, we will theoretically determine the potential maximum production rate of ion pairs by ionising radiation, based on our ionising radiation measurements and an assumed average energy expenditure of 34 eV for creating an ion pair, which is termed as the ionising capacity. The ionising capacity can be viewed as a measure of the theoretical maximum ionisation rate, which however, may not well capture the true ionisation rate due to uncertainties in ionising radiation measurements, possible energy dissipation of ionising radiation in excitation and the invalidity concern associated with the use of 34 eV per production of an ion pair at near ground level in our calculation.'

12. P5 L15 "descripted" -> "described"

**Thank you for spotting out this typo! It is corrected in the revised manuscript.**

13. P6 L17 All ion spectrometers are defined by their upper and lower mobility limits, so this sentence is meaningless.

**The reviewer is right. The sentence is removed from the revised manuscript.**

14. Why is the upper mobility limit 3.2 cm-2V-1s-1?

**The upper mobility limit of the instrument is determined by its configurational design, hardware limitations and transmission optimisation.**

15. P6 L20 What are the potential consequences of the choice of mobility-diameter relationship and its parameters?

**Stokes-Millikan equation is a widely-accepted model for describing the relationship between electric mobility and diameter and the mobility diameter is defined based on this model. However, it has a deficiency in characterising the connection of diameter to mobility in the microscopic limit. Scientists, such as Tammet (1995) and Li and Wang (2003), have sourced the first approximation of Chapman-Enskog theory for extending the model to free molecular regime. However, they used different function forms of potentials to treat the interactions between a particle and the gas medium. In addition, also the choice of the coefficients in calculating the Cunningham slip correction factor could bring bias in the converted diameter from electric mobility (Kim et al., 2005). Moreover, the relationship could also be affected by variations in temperatures and densities of particles and the gas medium. Ehn et al. (2011) made a comparison between mobility, mobility diameter and mass diameter using different models and particle densities, as shown in the figure below. In general, the relationship between electric mobility and mass diameter agree well with those experimentally determined values. Although in this comparison, Ehn et al. did not take into account the mass correction factor proposed by Tammet (1996) in converting the electric mobility into mobility diameter, by simply using the relationship between mass diameter (dm) and mobility diameter (dp), dp = dm+0.3 nm suggested by Ku and Fernández de la Mora (2009), some clues on the possible changes in mobility diameters can be deduced.**

[Figure]

16. P7 L1-2 please be consistent with units.

**It was our carelessness. The form 'l/min' is used in all places in the revised manuscript.**

17. P7 L2 how and why is only one eighth of the air sampled?

**The instrument was designed by Hannes Tammet from the University of Tartu. A detailed description of the device can be found in Tammet (2006). Briefly, the device consists of two aspiration type plate condensers and each one is divided into 8 sections by plate electrofilters. Ions are only allowed to enter the condenser through the two central sections. However, since only the half of the condenser is used for ion collection, as shown in the figure below (taken from Tammet (2006)), therefore only one eighth of the air sucked in is considered as sample flow. The other half of the condenser that is not in use for ion collection serves the insulation purpose and allows the optimisation in transfer function determination.**

[Figure]

18. P7 L13 "programme" -> "program"

**Thank you for pointing out this! It is corrected now.**

19. P8 L8 Please explain what the condensation sink is and its units; not everyone will be familiar with it.

**The definition of condensation sink (CS) is added into the revised manuscript, as follows**

**'CS accounts for the loss rate of vapours due to condensational uptake by aerosol particles in the atmosphere (Kulmala et al., 2001).'**

20. P8 L6-18 this is quite a lot of meteorological detail for work which is presumably published elsewhere. Could you shorten it and cite a reference?

**Some of measurement details can be found from published research articles and some from the station's home page. We shortened this section in the revised manuscript as the referee suggested as follows**

**'The snow cover depth was measured manually on a weekly basis on seven different locations at the SMEAR II station. Measurements on soil temperature and soil volumetric water content were described by Pumpanen et al. (2003) and Ilvesniemi et al. (2010). Only the organic horizon data (5 cm depth, above the mineral layer (Pumpanen et al., 2003)) were used in this work. The organic horizon is in direct contact with the atmosphere, the condition of which exerts the primary influence on radon exhalation. The ambient relative humidity and air temperature data were taken from the mast measurement at 16 m and 4.2 m, respectively. More detailed description of the mast instrumentation can be found from the home page of the measurement site (http://www.atm.helsinki.fi/SMEAR/index.php/smear-ii/measurements).'**

21. P8 L22 Notwithstanding the comments above, should these units include a "per unit volume"?

**The referee is right. We corrected the definition of the ionising capacity in the revised manuscript as follows**

'The ionising capacity (*Q*) is defined as the potential amount of ion pairs produced per unit time in a unit volume upon ionisation by ionising radiation in the atmosphere.'

22. P9 L15 This statement is meaningless since you derive the "ionising capacity" directly from dose rate in equation 1. Please remove

**We used the measured ambient temperature and pressure in converting the dose rate to the ionising capacity according to eq. 1. Therefore, strictly speaking, there are small differences between the variations of ionsing radiation and the derived ionising capacity. For this reason, we would like to keep this sentence in the manuscript.**

23. P10 L19 Can you explain why marine air masses will have a low radon content; again, not all readers will necessarily know why.

**Thank you for the comment! As we have explained in the introduction of this research article, radon is the decay product of radium. Radium is much more abundant in rocks and soil than in water. We added such information in the revised manuscript:**

**'···marine air masses from Arctic and north Atlantic oceans, which favour NPF (Nilsson et al., 2001), typically have a low radon content (Chen et al., 2016). Radon comes from the radioactive decay of radium. Since marine surface water has a significantly lower radium content than the continental surface layer (Wilkening and Clements, 1975), only minor amount of radon can be collected by air masses traversing over the ocean.'**

24. P12 L10 "exam" -> "examine"

**Thank you pointing this out! The correction is made in the revised manuscript.**

25. P12 section 3.2.2 Since radon emits gammas, which are included in your gamma radiation as you said earlier, can you state more clearly that you only mean alphas and betas from radon here?

**The statement is added in the revised manuscript as follows**

'In comparison with the radon ionising capacity accounting for alpha and beta emissions of radon decay, the gamma ionising capacity exhibits a simpler pattern.'

26. P14 L8 would be better to say "growth season"

**Thank you for the suggestion! We changed it to 'growth season' in the revised version.**

27. P14 L23 not sure what is meant by, "there is a hindrance"

**Our apology for the unclearness. We reformulated the sentence as follows in the revised manuscript**

**'In the latter case, certain removal processes of ions from the cluster size range are inhibited, which can be either the growth of cluster ions to sizes bigger than 1.7 nm or the loss of cluster ions by the attachment to bigger particles.'**

28. P14 L24 what is the significance of the 1.7 nm threshold?

[Figure]

**The above figure shows a spectrum measured by the BSMA. It can be seen that on average, most ions are smaller than about 1.7 nm. Bigger ions are seen during the new particle formation event around noon. The critical cluster size was found to be 1.5±0.3 nm for atmospheric nucleation (Kulmala et al., 2013), after which further growth of a cluster in size is energetically favoured (Vehkamäki, 2006). Ions with a diameter larger than the critical**

cluster size can therefore be considered as nanoparticles. In view of BSMA spectra, 1.7 nm seems to be a good approximation of the critical cluster size as there is the constant presence of ions smaller than 1.7 nm.

29. P15 L14-18 As it stands, this is irrelevant to the rest of the paper and should be deleted.

**Thank you for the comment! However, since we suggested that both changes in atmospheric conditions and vapour sources could result in the dissimilar autumn and spring patterns in the cluster ion concentration, we would like to briefly introduce what kind of differences can be expected from vapour sources. For this reason, we consider information presented on these lines are of relevance.**

30. P15 L20-27 It is well known that negative atmospheric ions are smaller than positive ones, though this work offers more detail. Please cite the classical literature here.

**Thank you for the comment! We elaborated the discussion and added citations to the classical literature.**

**'While the positive polarity dominated the overall cluster ion concentration, more negative ions were seen in the first two sub-size ranges (0.8-1 nm and 1-1.2 nm). The former results from the electrode effect of the negatively charged earth surface, which repels negative ions in its vicinity; and it is a well-known phenomenon to the atmospheric electricity community (Israël, 1970;Wilson, 1921;Tinsley, 2008;Harrison and Carslaw, 2003). The latter agrees with observations that generally negative ions possess higher mean mobility than positive ions (Israël, 1970;Dhanorkar and Kamra, 1992;Hõrrak, 2001), i.e. on average, negative ions are of smaller sizes than positive ions.'**

31. P16 L16 not sure what is meant by molecular ions here.

**Ionising radiation produces ion pairs by ionisation. These ion pairs are known as primary ions, which contain electrons and some small molecular ions (Fig. 1 in the manuscript). These primary ions will undergo a series of physical collisions and chemical reactions with other constituents in the air, leading to the formation of more complex molecular ions, cluster ions, larger ions and neutral species. The similar variations seen in the 0.8-1 nm ion**

concentration to those in the ionising capacity could imply that the majority in this size range are probably molecular ions, which have not been heavily influenced by dynamical processes of cluster formation and therefore retained some features of primary ions.

'The 0.8-1 nm ion concentration showed features similar to those in the ionising capacity (Figs. 9 and 3), being high in the morning and low in the afternoon. This observation possibly indicates that the dominant population in the size range of 0.8-1 nm are molecular ions, which have not been heavily influenced by dynamical processes of cluster formation and therefore retained the characteristics of primary ions.'

32. P17 L4-6 This statement is speculative and needs to be qualified as such.

We agree with the referee that this statement of ours is speculative. However, qualification of it would require massive experimental work under carefully controlled conditions, which we consider are beyond the scope of this manuscript. We modified the text so that the message that this statement is a speculation and further work is needed for the verification is conveyed.

'This observation might be related to the proton affinity of water molecules ($H_2O$), which assigns $H_2O$ the ability to bind positive charges. The formed cations may constitute a portion of hydronium ions ($H_3O^+$), which are too small to be detected by the BSMA, resulting in the flattening-out of the 0.8-1 nm ion concentration. Yet, further experimental investigations are needed for the verification of this mechanism and quantification of its significance.'

33. P17 L7 You need to add some words on CS here.

We added explanation for the relation of CS to 0.8-1 nm ion concentrations in the revised manuscript as follows

'As the CS can provide a measure of the condensational loss rate onto aerosol particles, the relatively high CS, as seen in Figs. S2 e and f, could be an additional reason for the lower 0.8-1 nm ion production at high radon ionising capacities.'

34. P17 L20 what do you mean by "unaltered gamma ionising capacity level"?

By 'unaltered gamma ionising capacity level', we intended to describe the effect of temperature on the relationship between the 0.8-1 nm ion concentration and the gamma ionising capacity. As can be seen in Fig. 10 e and f, 0.8-1 nm ion concentrations tend to decrease with an increase in temperature at a given gamma ionising capacity.

The sentence is revised as follows

'When the $T$ was below 5 °C, the 0.8-1 nm ion concentration showed a linear relation with the gamma ionising capacity. In addition, more 0.8-1 nm ions also tended to appear at lower temperatures, similar to the temperature effect seen on the relationship between the 0.8-1 nm ion concentration and the radon ionising capacity.'

35. P19 L1 what do you mean by "clustering or simple charge binding?"

Certain vapours could cluster among themselves around a charge to save the charge from recombination. There are also vapours that have strong proton or electron affinity, which can steal charges from the original carriers and bind the charges on themselves. These are explained in the next 6 sentences in the same paragraph following 'This observation may be attributed to the production of certain vapours that compete with the recombination process and other sink mechanisms for electric charges either via clustering or simple charge binding'.

'This observation may be attributed to the production of certain vapours that compete with the recombination process and other sink mechanisms for electric charges either via clustering or simple charge binding. Ionising radiation can potentially free a large number of electric charges, which would 'sacrifice' themselves mostly in recombination, if not otherwise become detectable air ions. The survived electric charges take part in the formation of 0.8-1 nm ions mainly in the form of primary ions and molecular ions. Certain vapours can cluster among themselves around primary ions to form 0.8-1 nm ions. Charge-binding vapours, however, are able to take over charges from primary ions to form molecular ions. This charge transfer process may also be accompanied by chemical reactions between the vapour molecules and primary ions. Some of the molecular ions are possibly born with a size falling in the 0.8-1 nm size range.'

36. P19 L20  "focalised"  ->  "localised"

**Thank you for the suggestion! It is corrected in the revised manuscript.**

37. Table 3: The labels are not adequately explained – could perhaps talk about quantiles to make it clearer what you mean. Please also define STD. As discussed above, it would be better to talk about ionisation rates.

**We elaborated the explanation of the labels in the caption and used 25$^{th}$, 50$^{th}$ and 75$^{th}$ percentiles as notations. The revised table 3 as below**

**Table 1. Energy in eV m$^{-3}$ s$^{-1}$ deposited in the air by total environmental gamma radiation ($E_\gamma$) and by alpha and beta activities from radon-222 decay ($E_{Rn}$) based on the 2003-2006 data. The gamma and radon ionising capacities ($Q_\gamma$ and $Q_{Rn}$) in cm$^{-3}$ s$^{-1}$ were derived from the deposited energy assuming 34 eV for the generation of one ion pair in the air. The statistical features of these data are presented by the five-number summary with two additional measures, mean and standard deviation (STD).**

|  | **Min** | **25$^{th}$ percentile** | **50$^{th}$ percentile** | **75$^{th}$ percentile** | **Max** | **Mean** | **STD** |
|---|---|---|---|---|---|---|---|
| $E_\gamma$ | 0.769 | 242 | 314 | 338 | 544 | 295 | 72.3 |
| $E_{Rn}$ (α&β) | 0.145 | 28.6 | 50.3 | 79.5 | 284 | 58.1 | 38.8 |
| $Q_\gamma$ | 4.34 | 6.70 | 9.09 | 9.34 | 11.6 | 8.14 | 1.80 |
| $Q_{Rn}$ | 0.00427 | 0.84 | 1.48 | 2.34 | 8.35 | 1.71 | 1.14 |

38. Figure 2: It would be helpful to provide some indication of the variability, for example by a shaded band around the median.

**We appreciate the reviewer's comment. We added a shaded band around**

moving means to show the variability. The modified figure 2 is as follows

[Figure]

**Figure 2. Seasonal patterns of radon and gamma ionising capacities as a function of day-of-year over the years 2003-2006. The radon ionising capacity was determined from the alpha and beta radioactivity associated with radon-222 decay and the gamma ionising capacity from total gamma radiation. The data are presented as daily medians. Shaded bands are dedicated to outline the variabilities, expressed by standard deviations, of the data around their means determined using the moving average method.**

39. Figure 6: I don't understand where soil temperatures fit in here, is this a typo?

No, this is not a typo. Soil temperatures are shown on the x-axis in Fig. 6a. We apologise that the dashed line for section selection was laid on top of the x-label in Fig. 6a. It is dragged off the label now in the revised version. And as can be seen in Fig. 6a, the radon ionising capacity tended to increase with the elevated soil temperatures.

[Figure]

40. Figure 7: R2 needs a superscript on the 2

Thank you for spotting this out! It is corrected in the revised manuscript.

41. Figure 11: This plot tries to convey too much information. Would it be better to present all the data together (e.g. as a median), or pick one representative year and add the rest to the supplementary information?

**Thank you for the comment! Indeed, the plot contains too much information. Here we intended to deliver the message that we could see similar diurnal behaviours of 0.8-1 nm ions, of both polarities and in different seasons, in response to variations in the ionising capacity and boundary layer dynamics; the latter has a close relation to solar radiation. Since the two polarities show similar features, we will show only one polarity (neg.) in the main article and move the other polarity (pos.) to the supplement.**

**The revised Fig. 11 will be as follows**

[Figure]

**Figure 11. Diurnal patterns in median 0.8-1 nm negative ion concentrations, ionising capacities, global and UVB radiation intensities as well as modelled mixing layer heights (MLH) in different seasons over 2003-2006.**

**And the rest will be included in Fig. S4 as follows**

[Figure]

**Figure S4. Diurnal patterns in median 0.8-1 nm positive ion concentrations, ionising capacities, global and UVB radiation intensities as well as modelled mixing layer heights (MLH) in different seasons over 2003-2006.**

42. Figure 13: A linear fit may not be appropriate here given the theory – please revise

**The reviewer is right. As it is not a linear relationship between the ionising capacity and air ion concentration, the Pearson's correlation coefficient measure is not appropriate. We replaced them by Spearman's rho coefficients in the revised figure. The Spearman's rank correlation coefficient measures the dependency of the cluster ion concertation on the ionising capacity without requirements on the linearity or knowledge about the distributions of the variables.**

[Figure]

**Figure 13. The cluster (0.8-1.7 nm) ion concentration as a function of the ionising capacity (radon ionising capacity + gamma ionising capacity) for selected data in the years 2003-2006, with Spearman's rank correlation coefficient ($\rho$). The data was constrained on a) & b) event days and c) & d) non-event days in September-December between 0:00 and 3:00 with the wind direction between 280° and 30°, while the condensation sink (CS) was below 0.002 s$^{-1}$. No dependence of the cluster ion concentration on the CS or hour-of-day was identified. Also the snow season was screened out. The negative polarity is shown in a) and c) and the positive in b) and d), with soil temperatures in the organic layer (-4-0 cm above the mineral soil) shown on the colour scale.**

cluster ion formation and growth due to ionisation, vapour condensation, recombination etc. would be more instructive.

We appreciate the comment and suggestion of the reviewer. We added processes involved in cluster ion formation to Fig. 1. The original information on the demonstration of what are primary ions, molecular ions and cluster ions contained in Fig. 1 is retained. We intended to present clearly the relationship between primary ion, molecular ions and clusters ions and we think that a graphical illustration can better convey the message. We think that the suggestion of the reviewer would be a good addition to the figure and these processes are added now to describe better the underlying processes.

[Figure]

Figure 1. A schematic demonstration on the relationship between primary ions, molecular ions and cluster ions, as well as processes governing their formation and loss.

7. Section 3.1 and Figures 2-4 show seasonal analysis, splitting the annual cycle into 4 3-month periods. However, looking at Figure 2, the ' Spring' period shows a significant change in, in particular, the gamma ionising capacity. So, for example while the average ionising capacity (gamma + Rn) is similar to Winter when taken across this period March-May as a whole (Figure 3), the ionising capacity increases significantly during this period and also the relative importance of gamma and Rn changes. It may be worth repeating some of the analyses only for, say, mid-February to end-March, to examine this specific period where Rn contributes the largest fraction to overall ionising capacity across the whole year, as I wonder whether features are being diluted and missed by taking the average over a period in which large changes in these parameters occur.

**Thank you for the comment! The feature was blurred slightly by the 3-month average window, as can be seen from the figures below. There was a weak diurnal cycle in the min-Feb to end-March period and the ratio of medians in this period to medians in spring and in winter are 1.07 and 1.09, respectively. The contribution by radon in this mid-Feb to end-March period fell in between those in spring and winter. These hidden features are not very remarkable and, therefore, the choice of the four 3-month periods can be still considered representative.**

[Figure]

[Figure]

8. P4 L25 Please clarify whether ' during 2003-2006' means whole-year data collection for these years, or if not, in which months data collection/use began/ended.

**Thank you for the comment! The data were from our long-term routine measurements. There were small gaps every now and then due to maintenance activities. The ion spectrometer was introduced in 2003 and therefore there were about 2-month data missing in the beginning of 2003. But otherwise, we had almost full coverage of data over these four years. For the ionising radiation measurements, we had almost full data coverage in 2005 and 2006 for radon data and in 2003 and 2006 for total gamma. We mainly had gaps in ionising radiation measurements in the latter half of year 2004. However, gathering up the four-year data, we have a very large dataset and have data for every single day-of-year. Since we focused mostly on the general trends shown by the medians of the data, some gaps in the data could hardly bias our interpretation.**

**Information about data coverages is added in the revised manuscript**

**In section 2.1**

**'For the study period of 2003-2006, the data availability for air ions, particles, radon and gamma was 91%, 99%, 71% and 76%, respectively, allowing the coverage of every single day-of-year by each dataset.'**

9. P6 I note also that APi-TOF may not provide direct information on the actual electrical mobility of atmospheric cluster ions, as encountered in environmental measurements using air ion spectrometers, because of the evaporation of clustering water molecules during the APi-TOF

measurement.

**The low pressure used in mass spectrometers could indeed cause loss of water and other loosely attached molecules on the sampled clusters. Such breaking-up of clusters is more likely to happen in mass spectrometers than in air ion spectrometers. APi-ToF is a mass spectrometer for study the mass distribution of atmospheric clusters, but it provides no direct information on mobility. Although some information about the mobility of the sample clusters can be obtained via a comparison with an ion spectrometer, like the work down by Ehn et al. (2011), the mass-mobility relationship has some dependency on the choice of model, as shown by Ehn et al.**

10. P7 Please state the voltage scan time for the BSMA in this work (it looks like it appears in the legend to Figure 8 but should be specified in the main text as well).

   **We added this information in the revised manuscript as follows**

   **'A full measurement cycle scanning through the mobility range for both polarities takes 10 min.'**

11. P8 L11 Please clarify your justification for using only 'O horizon' soil data.

   **We add the reasoning for using O horizon soil data in the revised manuscript as follows**

   **'Measurements on soil temperature and soil volumetric water content were described by Pumpanen et al. (2003) and Ilvesniemi et al. (2010). Only the organic horizon data (5 cm depth, above the mineral layer (Pumpanen et al., 2003)) were used in this work. The organic horizon is in direct contact with the atmosphere, the condition of which exerts the primary influence on radon exhalation.'**

12. P4 L25 "Monitoring devices for" not "The monitoring devices of the"

   **Thank you for the correction! Changes are made accordingly**

13. P5 L15 "described" not "descripted"

**Thank you for spotting out our typo! It is corrected in the revised manuscript.**

14. P8 L8 "Time-Domain Reflectometer" or "...Reflectometry (TDR) device."

**Thank you for pointing this out! However, since the instrumentation for soil measurement has been described carefully by Pumpanen et al. (2003) and Ilvesniemi et al. (2010), we decided to remove the details from our paper, as suggested by the other reviewer. We add citations to those published works for readers who would like to know more about the measurement.**

15. P12 L10 "examine" not "exam"

**Thank you for the correction! It is changed accordingly.**

16. P12 L22 what is meant by "shading the variability"?

**We intended to say that we investigated the connection between the radon ionising capacity and RH after eliminating the influence brought by MLH, temperature and wind on the radon ionising capacity. And it is rephrased in the revised manuscript as follows**

**'··· no clear correlation between the radon ionising capacity and RH was found based on our dataset after ruling out the variability in the radon ionising capacity brought by MLH, temperature and wind.'**

17. P13 L9-11 move text "to the measured snow depth data" to L9: "exponential fitting ... could represent" or otherwise reword, to clarify what is relationship is being probed.

**Thank you for pointing out this! We reformulated the sentence as follows**

**'···the constant term in the exponential fitting to the measured snow depth data, being about 3 $cm^{-3}$ $s^{-1}$, could represent an approximation of the contribution by cosmic radiation to the ionising capacity.'**

18. P14 L9 "blooming" not "booming"? P20 L9 "localised" not "focalised"? P20 L26 "level out" not "level up" P22 L2 "support" not "supports"

**Thank you for these suggestions! Followings are the changes made in the revised manuscript: 'booming season' to 'growth season' , 'focalised' to 'localised' , 'level up' to 'level out' and 'support' to 'supports' .**

19. P22 L19-22 the full citation now appears to be available online, please amend.

**Thank you for pointing this out! The full citation is added in the revised manuscript.**

20. P33 L4 "0-4 cm" not "-4-0 cm". P34 Fig 2, both y-axes "ionising" not "ioning"

**Thank you for spotting these typos out! They are corrected accordingly.**

21. P43 L3 "T was below" not "T below"

**Thank you! It is corrected accordingly.**

22.

Track-change manuscript

**How do air ions reflect variations in ionising radiation in the lower atmosphere in a boreal forest?**

[revised manuscript text omitted]

Air ions were historically concerned in the discipline of atmospheric electricity (Israël, 1970), because their flow in the electric field of the earth-atmosphere system serves as the measureable conduction current in the atmosphere (Wilson, 1921;Tinsley, 2008;Harrison and Carslaw, 2003). The interest in atmospheric electricity could be traced back to the early 18ᵗʰ century when thunderstorms were suggested to be electrical phenomena (Herbert, 1997). However, only after Benjamin Franklin proposed the idea to draw electricity down from lightning in 1752, this theory was confirmed and the study of atmospheric electricity became popular (Herbert, 1997;Tinsley, 2008). Early efforts in this field were substantially invested into understanding lightning and electrification of clouds (e.g. (Canton, 1753b, a;Franklin, 1751)), even though there were reports on observations of atmospheric electricity under fair weather conditions (Canton, 1753b;Read, 1792;Bennett and Harrison, 2007). But why the air is conductive could not be explained. Meanwhile, Charles-Augustin de Coulomb observed gradual discharge of a well-insulated electroscope around 1785 and he attributed his observation to the contact of suspending particles present in the air (Walter, 2012;Angelis, 2014). This phenomenon was reproduced by Michael Faraday half a century later in 1835 (Angelis, 2014). Thanks to the further improvement of the electroscope by William Thomson and Lord Kelvin (Angelis, 2014;Flagan, 1998), Crookes (1878) could find that the discharge rate of an electroscope decreases with a decreasing air pressure, suggesting that it is the air inside the instrument that manipulates the discharge. Yet, the reasoning remained undisclosed, until the discovery of radioactivity by Wilhelm Röntgen, Henri Becquerel and Marie and Pierre Curie enabled Julius Elster and Hans Geitel from Germany and Charles Thomson Rees Wilson from Scotland to relate the spontaneous discharge of the electroscope to ionisation of the air by radioactive sources (Angelis, 2014;Carlson and De Angelis, 2011). Therefrom, the importance of air ions in the atmosphere emerged. Contemporaneously, the interest of Joseph John Thomson, director of the Cavendish Laboratory, in the charge carriers produced by ionising radiation motivated the development of instrumentations for measuring electrical charges in gases, leading to various valuable outcomes, e.g. the cloud chamber designed by C. T. R. Wilson, as well as techniques for measuring ion mobility by Ernest Rutherford and John Zeleny and for studying gaseous ion diffusion by John Sealy Townsend (Flagan, 1998;Robotti, 2006). These works laid the theoretical and instrumental foundation for later aerosol studies. The experimental results from C. T. R. Wilson's cloud

chamber measurements in 1895 and 1899 on the influence of ionising radiation on the formation of cloud droplets brought interests of air ions into the atmospheric aerosol community. Inspired by these early works, advancements in atmospheric aerosol studies progressed both instrumentally and theoretically over the century (e.g. (Tammet, 1970;Nolan, 1924;Hogg, 1939;Hewitt, 1957;Tammet, 2006;Reischl, 1991;Aplin and Harrison, 2000;Mason and McDaniel, 1988;Millikan, 1923;Hinds, 1999;Tammet, 1995;Rosell-Llompart and Mora, 1993)).

Devices employed for ion studies comprise different types of aspiration condensers, ion mobility spectrometers (IMS) and mass spectrometers (Hirsikko et al., 2011;Tammet, 1970;Laskin et al., 2012;Cumeras et al., 2015). Notably, modern key instruments for field observations of air ions are mainly aspiration contender-based devices and mass spectrometers, such as the Gerdien counter - an integral aspiration condenser (Gerdien, 1905;Aplin and Harrison, 2000;Vojtek et al., 2006), ion spectrometers designed by Airel Ltd – single or multiple channel aspiration condensers (Tammet, 2011, 2006;Mirme et al., 2007;Manninen et al., 2009;Kulmala et al., 2016) and the atmospheric pressure interface time-of-flight mass spectrometer (APi-ToF) (Junninen et al., 2010). While aspiration condensers provide information on the concentration and mobility of charge carriers, mass spectrometers reveal mainly the chemical properties of them. The IMS, however, has a limited application in studying ambient ions, due to difficulties in spectrum interpretation (Hirsikko et al., 2011). The purpose of these instrumentations is not only limited to air ion or air conductivity observations, but also for nano-material synthesis (Kruis et al., 1998) and for the improvement of the fundamental understanding on the relationship between mobility, mass and size (Ku and de la Mora, 2009).

At present, the number size distribution of air ions can be measured down to about 0.8 nm in Millikan mobility size by using ion spectrometers (Tammet, 2011;Manninen et al., 2009;Tammet, 2006). Most air ions concentrate at the lowest size band with a diameter of 0.8-1.7 nm (Manninen et al., 2009), which is generally known as the cluster size range (Tammet, 2012, 1995). In principle, this size range contains large molecular ions and clusters of molecular ions. Ions smaller than 0.8 nm comprise mostly

relatively simple molecular ions, which are either primary ions or originate from the survived fraction of primary ions from recombination. The critical cluster size was found to be 1.5±0.3 nm for atmospheric nucleation (Kulmala et al., 2013). Once the critical cluster size is reached, further growth of a cluster in size is energetically favoured (Vehkamäki, 2006). Air ions that have sizes larger than the critical cluster size are therefore typically viewed as nanoparticles, observable especially during new particle formation (NPF) events (Manninen et al., 2009;Kulmala et al., 2012). However, since the atmosphere is a vast pool full of clusters with different structures and compositions, the critical cluster size is rather a size range than a single size, as demonstrated in Fig. 1. As a consequence, there exists no such a clear size separation between cluster ions and charged nanoparticles, neither does it exist between molecular ions and cluster ions. Therefrom, although the size range of 0.8-1.7 nm may contain molecular ions, cluster ions, and even charged nanoparticles, hereafter we refer ions in this size range as cluster ions, unless otherwise mentioned.

Although it is known that ionising radiation creates ion pairs via ionisation in the atmosphere (Flagan, 1998;Harrison and Carslaw, 2003;Israël, 1970), except for a few attempts (Hirsikko et al., 2007;Laakso et al., 2004), minor efforts have been invested in understanding the connection between ionising radiation and observed air ions in the lower atmosphere. Moreover, there is a lack of quantification on the underlying processes. Such deficiencies prompt the motivation of this work to examine how variations in ionising radiation are reflected in observed air ions based on ambient measurement. The aims of this study are 1) to identify the key factors responsible for the variability in ionising radiation and in observed air ion concentrations, 2) to reveal the linkage of observed air ions to the variations in ionising radiation and 3) to provide an in-depth analysis on the effects of ionising radiation on air ion formation. We will first introduce factors that cause the seasonal and diurnal variability in ionising radiation and air ions, and then exposit the connection of observed air ions to variations in ionising radiation and the influence of different atmospheric conditions on this relationship. To assist our analysis, we will theoretically determine the potential maximum production rate of ion pairs by ionising radiation, based on our ionising radiation measurements and an assumed average energy expenditure of 34 eV for creating an ion pair, which is termed as the ionising capacity. The ionising capacity can be

viewed as a measure of the theoretical maximum ionisation rate, which however, may not well capture the true ionisation rate due to uncertainties in ionising radiation measurements, possible energy dissipation of ionising radiation in excitation and the invalidity concern associated with the use of 34 eV per production of an ion pair at near ground level in our calculation.

**2. Materials and methods**

[revised manuscript text omitted]

25   in the atmosphere (Kulmala et al., 2001).

The snow cover depth was measured manually on a weekly basis on seven different locations at the SMEAR II station. Measurements on soil temperature and soil volumetric water content were described by Pumpanen et al. (2003) and Ilvesniemi et al. (2010). Only the organic horizon data (5 cm depth, above the mineral layer (Pumpanen et al., 2003)) were used in this work. The organic horizon is in direct contact with the atmosphere, the condition of which exerts the primary influence on radon exhalation. The ambient relative humidity and air temperature data were taken from the mast measurement at 16 m and 4.2 m, respectively. More detailed description of the mast instrumentation can be found from the home page of the measurement site (http://www.atm.helsinki.fi/SMEAR/index.php/smear-ii/measurements).

**2.2. Ionising capacity**

The ionising capacity ($Q$) is defined as the potential maximum amount of ion pairs produced per unit time in a unit volume upon ionisation by ionising radiation in the atmosphere. The ionising capacity in this work was determined on the basis of the assumption that 34 eV is needed for the production of an ion pair in the air. The energy released by radon-222 and its short-lived progeny during alpha and beta decay were taken into consideration. The contributions from actinon (radon-219) and thoron (radon-220) were excluded in our study, because they are present in trace amounts naturally and have remarkably shorter half-lives (3.96 s for actinon and 55.6 s for thoron) than radon-222, which hardly permit them enough time to migrate out of the ground, especially in the case of actinon. Frozen ground and snow cover could substantially cease the transportation of thoron to the atmosphere during cold months. Even the vegetation reduces the flux of thoron from the ground to the surface air (Mattsson et al., 1996). Besides, Laakso et al. (2004) found only little contribution by thoron in the SMEAR II station to the overall radon activity. For radon-222, the decay mode and energy accounted for the ionising capacity conversion are listed in Table 1. The minor gamma fraction from the radon-222 decay was assumed to be detectable by the gamma spectrometer. Therefore, the data obtained by the gamma spectrometer could be considered to represent the total gamma radiation, including the terrestrial fraction, the cosmic

fraction and the fraction from radon decay. In addition, the gamma spectrometer also accounts ionising energy from muons.

[revised manuscript text omitted]

during the actual clustering process from both primary ions and molecular ions, and c) after clustering via the charge uptake from both primary ions and molecular ions. Recombination consumes part of these acquired charges, leaving the rest retained eventually in the form of cluster ions. In the latter case, certain removal processes of ions from the cluster size range are inhibited, which can be either the growth of cluster ions to sizes bigger than 1.7 nm or the loss of cluster ions by the attachment to bigger particles. All of these mechanisms, however, are manipulated primarily by atmospheric conditions. Atmospheric conditions, such as the temperature and relative humidity, can directly influence the rate of clustering and growth. Nonetheless, they also modify atmospheric compositions via the production of functional vapours involved in clustering or growth and via altering the availability of precursor gases of these vapours. As a consequence, these phenomena, seen in Fig. 8, are likely brought by a synergy of complicated atmospheric dynamic processes.

Although high ionising capacities were found in the morning, on average the morning cluster ion concentration was not so high as the evening level during the relatively warm months between April and October (Fig. 8 a). Especially in autumn months, the enhancement in air ion production from the high morning ionising capacity was not reflected in the cluster ion concentration. Such observations may result from the cluster formation process becoming inferior to the preferred particle growth due to photochemical processes, while facing the dilution led by the expansion of the mixing volume. Autumn was the second peak period for the occurrence of NPF events at the SMEAR II station after spring (Nieminen et al., 2014). The dissimilar autumn and spring patterns in the cluster ion concentration originate likely from differences in atmospheric conditions and vapour sources. For example, spring UVA radiation intensities are higher than the autumn ones, while the RH shows an opposite behaviour (Lyubovtseva et al., 2005). Biogenic VOC emissions have a strong seasonality (Tarvainen et al., 2005;Hakola et al., 2012), which is reflected in their atmospheric concentrations. Hakola et al. (2012) demonstrated that monoterpenes tend to dominate VOCs in late summer and autumn, while aromatic hydrocarbons dominate in spring and early summer. Sesquiterpene emissions and concentrations were found to be high in late summer and autumn (Hakola et al., 2012;Tarvainen et al., 2005, 2007).

**3.3.2. Variations in cluster ion concentrations in sub-size ranges**

Ion concentrations in different sub-size ranges (0.8-1 nm, 1-1.2 nm and 1.2-1.7 nm) of cluster sizes showed distinct patterns (Fig. 9). While the positive polarity dominated the overall cluster ion concentration, more negative ions were seen in the first two sub-size ranges (0.8-1 nm and 1-1.2 nm). The former results from the electrode effect of the negatively charged earth surface, which repels negative ions in its vicinity; and it is a well-known phenomenon to the atmospheric electricity community (Israël, 1970;Wilson, 1921;Tinsley, 2008;Harrison and Carslaw, 2003). The latter agrees with observations that generally negative ions possess higher mean mobility than positive ions (Israël, 1970;Dhanorkar and Kamra, 1992;Hõrrak, 2001), i.e. on average, negative ions are of smaller sizes than positive ions. 0.8-1 nm ion concentrations were found to be the lowest (around 100 cm$^{-3}$) in all seasons. There was a difference of 150-200 cm$^{-3}$ between the concentrations of 0.8-1 nm ions and 1-1.2 nm ions through all seasons in both polarities, with a larger difference observed in summer and smaller in winter. A more pronounced seasonality was observed for 1.2-1.7 nm ions, with the summer concentrations being the highest and winter concentrations being the lowest.

[revised manuscript text omitted]

After the mixing layer had fully developed in spring, summer and autumn, there could be observed a transient boost in the 0.8-1 nm ion production (especially of the positive polarity) along with the

15  shrinkage of the mixing layer, even prior to a clear recovery of the ionising capacity (Fig. 11). This observation may be attributed to the production of certain vapours that compete with the recombination process and other sink mechanisms for electric charges either via clustering or simple charge binding. Ionising radiation can potentially free a large number of electric charges, which would 'sacrifice' themselves mostly in recombination, if not otherwise become detectable air ions. The survived electric

20  charges take part in the formation of 0.8-1 nm ions mainly in the form of primary ions and molecular ions. Certain vapours can cluster among themselves around primary ions to form 0.8-1 nm ions. Charge-binding vapours, however, are able to take over charges from primary ions to form molecular ions. This charge transfer process may occur either via charge exchange ionisation or via chemical reactions between the vapour molecules and primary ions. Some of the molecular ions are possibly born

[revised manuscript text omitted]

transformations to become cluster ions. Therefore, the linkage between the cluster ion concentration and ionising capacity was traceable statistically. However, this connection was very poorly preserved on non-event days, possibly because after being created, charges went through a too complicated series of modifications before reaching the cluster sizes. These observations may also provide some indirect measure of the relative duration of dynamic processes faced by newly formed charges before they become cluster ions under different atmospheric conditions: it might suggest that charges became cluster ions in a shorter time scale on NPF event days than on non-event days and as a consequence, the cluster ions on NPF event days are enabled to retain some features in connection to ionising radiation.

**4. Conclusions**

In this work, diurnal and seasonal cycles in ionising radiation were presented and key influencing factors responsible for these features were overviewed in order to investigate how observed air ions respond to these variations and to improve our understanding on air ion formation. To assist the analysis, a term, ionising capacity, was introduced to capture patterns in ionising radiation. The ionising capacity was determined theoretically as the potential maximum production rate of ion pairs in the atmosphere by ionising radiation, based on the assumption that an ion pair is produced upon every 34 eV energy dissipation of ionising radiation. The data used in this study were collected from ambient measurements during 2003-2006 from a boreal forest site in southern Finland. In our analysis, the accounted ionising radiation is composed of energy from alpha and beta decays of Rn-222 and accompanying gamma radiation, energy contained in the gamma radiation from terrestrial origins as well as gamma radiation and muons released from the interactions between cosmic rays and air molecules. However, a portion of high-energy cosmic ray muons may be missing from our scope. Variations in the ionising capacity were primarily related to boundary layer development, soil conditions, snow accumulation and the origin of air masses.

Although ionising radiation is known to be responsible for air ion production, patterns in the measured air ion concentration in the cluster size range (0.8-1.7 nm) did not exhibit a highly comparability to

those in the ionising capacity, due to modifications of air ion properties exerted by different dynamical processes and chemical reactions during the evolution of charges in the atmosphere. Nevertheless, the connection of air ions to ionising radiation was seen for air ions detected in the lowest detected size band (0.8-1 nm) of the cluster size range (0.8-1.7 nm). The evolvement of these 0.8-1 nm ions with time to larger sizes in the cluster size band was also identified, affirming the primary role of ionising radiation in the production of air ions in the lower atmosphere. Yet, atmospheric conditions, such as temperature, humidity and pre-existing aerosol particles, brought complications into this relationship. By carefully constraining data to conditions of a similar meteorology, seasonality, diurnality and amount of background aerosol particles, a strong dependency of total cluster ion concentrations on the ionising capacity was identified on new particle formation (NPF) days. However, the linkage was not visible on non-event days. These observations may suggest that charges, after being born, underwent different processes on NPF days and non-event days and possibly indicate also that the transformation of newly formed charges to cluster ions occurred faster on NPF days than on non-event days. These results could help to advance our understanding on the role of ions in atmospheric new particle formation.

However, to obtain further insights into the fate of charges created by ionising radiation in the atmosphere, i.e. ion balance, and into the role of air ions in the atmospheric new particle formation process, it is crucial to understand the transformation process of electric charges into detectable air ions. For this purpose, knowledge on the number size distribution of air ions smaller than 0.8 nm is of necessity. Additionally, theoretical understanding on the formation mechanisms of cluster ions from molecular ions needs to be deepened. Conjointly, also advancing instrumental development for the detection of sub-0.8 nm ions could be worthy of being brought onto the agenda.

**5. Acknowledgement**

This work received funding support from the Academy of Finland Centre of Excellence (project no. 272041 and 1118615), European Union's Horizon 2020 research and innovation programme under grant agreement No. 654109 (ACTRIS-2) as well as the European Union Seventh Framework Programme (FP7/2007-2013 ACTRIS) under grant agreement No. 262254. Also the CRyosphere-Atmosphere Interactions in a Changing arctic Climate (CRAICC) project of the Nordic Centre of Excellence is acknowledged. The authors appreciate the valuable communication with Prof. Jaana Bäck, Dr. Pasi Kolari, Dr. Anne Hirsikko and Dr. Juha Hatakka.

**Tables**

Table 1. Decay modes and energy of radon-222 and its short-lived progeny taken into account in the ionising capacity determination. Decay partitioning is accounted in defining the weighted average decay energy. The data are extracted from the National Nuclear Data Centre of Brookhaven National Laboratory (http://www.nndc.bnl.gov/chart/).

| Nuclide | Decay mode | Weighted average decay energy [keV] |
|---|---|---|
| $^{222}$Rn | α (100%) | 5589 |
| $^{218}$Po | α (99.98%) | 6001 |
| $^{214}$Pb | β⁻ (100%) | 225 |
| $^{214}$Bi | β⁻ (100%) | 642 |
| $^{214}$Po | α (100%) | 7687 |

Table 2. Median radon ionising capacity in $cm^{-3}$ $s^{-1}$ on new particle formation (NPF) event and non-event days in different seasons over 2003-2006.

| NPF classification | Event | Non-event |
|---|---|---|
| Spring | 0.68 | 1.31 |
| Summer | 0.90 | 1.74 |
| Autumn | 0.99 | 1.8 |
| Winter | 0.56 | 1.96 |

Table 3. Energy in eV m$^{-3}$ s$^{-1}$ deposited in the air by total environmental gamma radiation ($E_\gamma$) and by alpha and beta activities from radon-222 decay ($E_{Rn}$) based on the 2003-2006 data. The gamma and radon ionising capacities ($Q_\gamma$ and $Q_{Rn}$) in cm$^{-3}$ s$^{-1}$ were derived from the deposited energy assuming 34 eV for the generation of one ion pair in the air. The statistical features of these data are presented by the five-number summary with two additional measures, the mean and standard deviation (STD).

|  | Min | 25$^{th}$ percentile | 50$^{th}$ percentile | 75$^{th}$ percentile | Max | Mean | STD |
|---|---|---|---|---|---|---|---|
| $E_\gamma$ | 0.769 | 242 | 314 | 338 | 544 | 295 | 72.3 |
| $E_{Rn}$ (α&β) | 0.145 | 28.6 | 50.3 | 79.5 | 284 | 58.1 | 38.8 |
| $Q_\gamma$ | 4.34 | 6.70 | 9.09 | 9.34 | 11.6 | 8.14 | 1.80 |
| $Q_{Rn}$ | 0.00427 | 0.84 | 1.48 | 2.34 | 8.35 | 1.71 | 1.14 |

**Figures**

Figure 1. A schematic demonstration of the relationship between primary ions, molecular ions and cluster ions, as well as processes governing their formation and loss.

[revised manuscript text omitted]

5   are presented as daily medians. Shaded bands are dedicated to outline the variabilities, expressed by standard deviations, of the data around their means determined using the moving average method.

[revised manuscript text omitted]